# LIFELONGAGENTBENCH :
# EVALUATING LLM AGENTS AS LIFELONG LEARNERS

## ABSTRACT

Lifelong learning is essential for intelligent agents operating in dynamic environments. Current large language model (LLM)-based agents, however, remain stateless and unable to accumulate or transfer knowledge over time. Existing benchmarks treat agents as static systems and fail to evaluate lifelong learning capabilities. We present LifelongAgentBench, the *first* unified benchmark designed to systematically assess the lifelong learning ability of LLM agents. It provides skill-grounded, interdependent tasks across three interactive environments—Database, Operating System, and Knowledge Graph—with automatic label verification, reproducibility, and modular extensibility. Extensive experiments reveal that conventional experience replay has limited effectiveness for LLM agents due to irrelevant information and context length constraints. We further introduce a *group self-consistency* mechanism that significantly improves lifelong learning performance. We hope LifelongAgentBench will advance the development of adaptive, memory-capable LLM agents. [1]

## 1 INTRODUCTION

The rapid development of large language models (LLMs) has revolutionized language-based artificial intelligence, achieving state-of-the-art performance across a wide range of natural language processing tasks. Recently, research has shifted from static models to LLM-based agents, designed to interact with dynamic environments, perform complex decision-making, and continuously improve through experience. These agents combine the language understanding and generation capabilities of pretrained LLMs with autonomous action selection and interaction policies.

However, a critical limitation remains: today's LLM-based agents fundamentally lack memory and the ability to incrementally accumulate knowledge over time. They operate in a **stateless manner**, treating each task independently without the capacity to remember, adapt, or transfer past experiences. Achieving *general artificial intelligence* demands agents that can **continuously acquire, retain, and reuse knowledge across diverse environments and long time horizons**. This lifelong learning capability is widely regarded as a cornerstone of human-level intelligence but remains largely unaddressed in current agent research (Zheng et al., 2025a;b; Li et al., 2025).

Existing LLM agent benchmarks (Zhou et al., 2024; Koh et al., 2024b; Liu et al., 2023) have been designed under the static agent paradigm. They focus on isolated tasks, ignoring inter-task dependencies, skill reuse, and the realistic challenges of knowledge retention and catastrophic forgetting. More critically, there is currently **no standardized benchmark for systematically evaluating lifelong learning in LLM agents**. This absence has severely limited progress toward developing agents capable of lifelong adaptation and memory. Additionally, practical adoption is hindered by label inaccuracies (Yang et al., 2025), lack of verifiability, and poor reproducibility in prior benchmarks.

To address these critical gaps, we propose LifelongAgentBench, the first unified benchmark specifically designed to evaluate the lifelong learning capabilities of LLM-based agents across realistic and diverse interactive environments (Figure 1). LifelongAgentBench systematically tests agents' abilities to acquire *atomic skills*, transfer them across tasks, and maintain stable performance over long

---

[1] Source code will be publicly available: `https://anonymous.4open.science/r/continual_agent_bench-4F3B`

Table 1: Comparison between LifelongAgentBench and existing benchmarks. †: Yang et al. (2025) highlights label error issues in WebArena.

| Benchmark | WebArena (Zhou et al., 2024) | VisualWebArena (Koh et al., 2024b) | AgentBench Liu et al. (2023) | VisualAgentBench (Liu et al., 2024) | LifelongAgentBench (Ours) |
|---|---|---|---|---|---|
| Task Execution Scheme | Parallel execution with undetermined execution order | | | | Serial execution with fixed order and historical dependency retention |
| Task Dependency | ✗ | | | | ✓ |
| Knowledge Transfer | ✗ | | | | ✓ |
| Modular Extension | ✗ | | | | ✓ |
| Supported Environment | Web Only | | Environment with serializable action and observation | | Environment with serializable action and observation |
| Label Verification Mechanism | Human Annotation † | | Automatic Label Verification | | Automatic label verification via human review, LLM judgement, and execution results |
| Deployment | Single Machine Deployment | | Distributed Deployment | | Both Single Machine and Distributed Deployment |
| Transparent RPC | / | | ✗ | | ✓ |
| Code Quality Control | Not declared | | Code are partially check by black, mypy, beartype. | | All code are checked by black and mypy (strict). |
| # Instance | 812 | 910 | 1091 | 746 | 1396 |

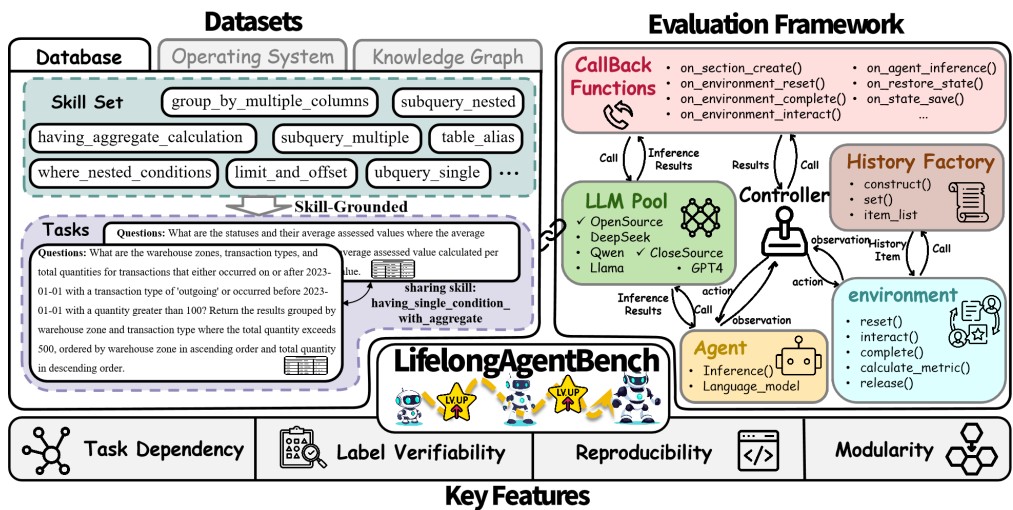

Figure 1: Overview of LifelongAgentBench: a unified dataset and evaluation framework with skill-grounded tasks, modular components, and four key properties: task dependency, label verifiability, reproducibility, and modularity.

sequences of dependent tasks. It includes three task-rich environments—Database (DB), Operating System (OS), and Knowledge Graph (KG)—to simulate complex, evolving scenarios requiring continuous learning.

LifelongAgentBench provides four key innovations that set it apart from existing LLM agent benchmarks: **Task Dependency:** Tasks are skill-grounded and explicitly designed to quantify inter-task relatedness, enabling rigorous analysis of knowledge transfer and catastrophic forgetting. **Label Verifiability:** Each environment includes automatic label verification (e.g., SQL query validation, OS state hashing, SPARQL output verification) to ensure objective and reproducible evaluation. **Reproducibility:** The benchmark provides a fully containerized infrastructure and modular design, making it straightforward for researchers to reproduce experiments and extend the framework. **Modularity:** The platform offers extensible callback functions and a pluggable LLM agent interface, supporting both open-source and commercial models such as LLaMA (Grattafiori et al., 2024), DeepSeek (Guo et al., 2025), Qwen (Yang et al., 2024), and GPT-5 (OpenAI, 2025). A detailed comparison between LifelongAgentBench and prior benchmarks is presented in Table 1.

We conduct extensive experiments using LifelongAgentBench, yielding several key insights: (1) While experience replay is effective in traditional continual learning, its impact in agent settings varies significantly depending on model size, architecture, and task complexity. (2) Increasing the volume of past experience does not always improve performance and can even degrade it due to irrelevant information and context length limitations. (3) To mitigate this, we propose a novel *group self-consistency* mechanism, which partitions historical experience into groups and applies voting strategies to improve decision quality. We show that group self-consistency significantly enhances the effectiveness of experience replay across multiple model backbones.

Our contributions are threefold: (1) We introduce LifelongAgentBench, the first unified benchmark specifically designed to evaluate the lifelong learning capabilities of LLM-based agents across diverse, realistic interactive environments. (2) We provide the first systematic analysis of lifelong

learning in LLM agents, revealing key limitations of conventional experience replay due to irrelevant information and context length constraints. (3) We propose a novel *group self-consistency* mechanism that partitions historical experiences and applies voting strategies, significantly enhancing the effectiveness of lifelong learning across multiple LLM backbones.

## 2 RELATED WORK

**Lifelong Learning.** Lifelong learning, or continual learning, aims to enable AI systems to acquire and retain knowledge across sequential tasks while mitigating catastrophic forgetting (French, 1999). Prior research has primarily focused on static, non-interactive settings such as image classification or continual instruction tuning, where models are fine-tuned on sequential datasets without interacting with an external environment (Zheng et al., 2025b;a). In these tasks, both inputs and outputs are fixed, and the model is not required to actively take actions or adapt based on environmental feedback. In contrast, lifelong learning for LLM-based agents interacting with complex environments over long horizons remains largely unexplored.

**LLM Agent Benchmarks.** Several benchmarks have been proposed to evaluate the capabilities of LLM-based agents. WebArena (Zhou et al., 2024), AgentBench (Liu et al., 2023), and Visual-WebArena (Koh et al., 2024b) offer valuable evaluation settings but focus on single-episode performance in static environments. These platforms lack mechanisms to model sequential decision making, cumulative learning, or skill transfer across tasks. While recent works have explored LLM agents in interactive scenarios such as game playing (Fan et al., 2022) and tool use (Qin et al., 2023), they do not provide standardized lifelong learning protocols.

LifelongAgentBench addresses this critical gap by providing the first benchmark specifically designed to evaluate LLM agents under lifelong learning constraints. It introduces reproducible lifelong evaluation with persistent environment states, explicit task dependencies via skill taxonomies, and scalable experience replay, establishing a foundation for systematic study of generalization, skill transfer, and long-term retention in LLM agents.

## 3 PROBLEM FORMULATION OF LIFELONG LEARNING FOR LLM AGENTS

We model lifelong learning for LLM-based agents as sequential decision making over a series of tasks, each framed as a goal-conditioned partially observable Markov decision process (POMDP) (Zheng et al., 2025b). **Environment:** An environment is $\mathcal{E} = (\mathcal{S}, \mathcal{A}, \mathcal{G}, T, R, \Omega, O)$, where $\mathcal{S}$ is the state space; $\mathcal{A}$, natural language actions; $\mathcal{G}$, task goals; $T$, state transitions; $R$, rewards; $\Omega$, observations; and $O$, the observation function. LifelongAgentBench provides DB, OS, and KG environments. **Agent and Task:** An LLM agent follows a policy $\pi : \Omega \rightarrow \mathcal{A}$ mapping observation $o_t$ to action $a_t$. A task is $\mathcal{T}^{(i)} = \langle \mathcal{E}^{(i)}, o_0^{(i)}, g^{(i)} \rangle$, with $o_0^{(i)}$ as the initial observation and $g^{(i)}$ as the goal. The agent generates a trajectory $\xi^{(i)} = (o_0, a_0, r_0, \ldots, o_T, a_T, r_T)$, receiving a single reward upon submitting a final answer (success = 1, failure = 0). **Objective:** Given tasks $\mathcal{U} = \{\mathcal{T}^{(1)}, \ldots, \mathcal{T}^{(n)}\}$, the goal is to maximize cumulative expected reward: $\max_\pi \sum_{i=1}^n \mathbb{E}_{\xi^{(i)} \sim \pi} \left[ \sum_{t=0}^T R(o_t, a_t, g^{(i)}) \right]$

LifelongAgentBench evaluates agents on their ability to leverage past experience to improve current task performance.

## 4 DATA CONSTRUCTION

To rigorously evaluate LLM agents in lifelong learning scenarios, we introduce a novel and meticulously constructed benchmark dataset composed of three distinct and challenging environments: Database, Operating System, and Knowledge Graph. Unlike conventional benchmarks that often rely on isolated, simplistic tasks with loosely defined inter-task relationships, our dataset is innovatively designed to reflect complex, realistic lifelong learning contexts. The key contributions of this dataset include the systematic generation of tasks explicitly tied to clearly defined *atomic skills*, sophisticated methodologies for controlling *skill distribution* and *task complexity*, and rigorous noise management to simulate real-world variability. The dataset's construction required extensive validation and curation efforts, underscoring the intricacy and robustness of our approach. Detailed

descriptions of the construction procedures and sample data are provided in Appendix A and Appendix C, respectively.

## 4.1 DESIGN PRINCIPLES

The data construction process follows three core principles. First, we adopt a skill-centric task generation approach. Each environment $\mathcal{E}^{(i)}$ is characterized by a set of atomic skills $\mathcal{SK}_{\mathcal{E}^{(i)}}$, where the number of skills $N_{\mathcal{E}^{(i)}}$ varies with the environment's complexity. Each task $\mathcal{T}_j^{(i)}$ is associated with a subset of these skills $\mathcal{SK}_{\mathcal{E}^{(i)}}^{(j)}$, ensuring consistent competency representation across tasks. The relationship between tasks $m$ and $n$ is quantified by the harmonic mean of shared skill proportions:

$$as_{\mathcal{E}^{(i)}}^{(m,n)} = 2as_{\mathcal{E}^{(i)}}^{(m)}as_{\mathcal{E}^{(i)}}^{(n)}/(as_{\mathcal{E}^{(i)}}^{(m)} + as_{\mathcal{E}^{(i)}}^{(n)}) \qquad (1)$$

where $as^{(m)}$ and $as^{(n)}$ denote the proportion of shared skills relative to each task's total skills. This formulation captures both commonality and uniqueness across tasks.

To mitigate skill isolation, we employ a probabilistic sampling strategy where infrequent skills have higher sampling probabilities, ensuring balanced representation across the dataset. Noise levels are controlled by regulating the proportion of tasks containing rare skills, facilitating robustness analysis. Tasks span simple, intermediate, and complex configurations to mimic real-world variability and allow evaluation across progressive difficulty levels. As shown in Figure 2, extensive connections exist between skills across tasks. A summarization of the skill set in each enviroment is in Table 5.

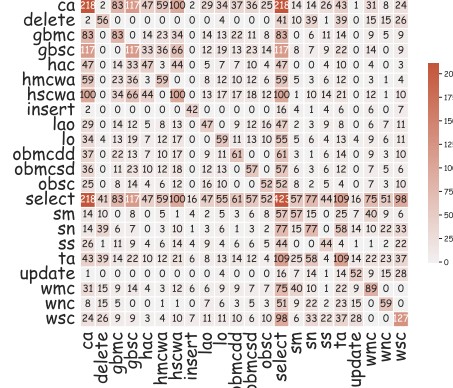

Figure 2: Skill concurrency in the Database environment.

## 4.2 ENVIRONMENT IMPLEMENTATIONS

**Database Environment:** We implement this environment using Docker-containerized MySQL instances to ensure reproducibility. A fresh MySQL container is created for each experimental run to maintain task isolation. Tasks are initialized by generating a database table with predefined attributes, which is deleted upon task completion. We identify 22 SQL-related skills, including column aliasing, complex filtering with WHERE and HAVING clauses, multi-column grouping, data manipulation (INSERT, UPDATE, DELETE), and nested subqueries. Detailed description of each skill is provided in Appendix A.1.1.

Task construction begins by sampling skills, with infrequent skills prioritized. SQL queries corresponding to sampled skills are generated using the DeepSeek-R1 model. Each query is executed on a synthetic database instance, and invalid or inconsistent tasks are discarded. To prevent skill imbalance, we require a minimum of 20 occurrences per skill across 500 selected tasks. Task correctness is verified both automatically (e.g., result matching, MD5 hashing of database states) and manually by inspecting 10% of randomly sampled tasks for syntax and logical coherence.

**Operating System Environment:** This environment leverages disposable Docker containers running Ubuntu to isolate tasks. Containers are destroyed and re-instantiated after each task. We define 29 Bash command skills, including file manipulation (cp, mv, rm), user management (useradd, groupadd), text processing (awk, grep, sed), and system monitoring (ps, top). Tasks are grouped by complexity: simple (1–4 commands), intermediate (5–8), and complex (9–12). Detailed description of each skill is provided in Appendix A.2.1.

Command sequences are generated using DeepSeek-R1, ensuring logical consistency across multiple steps. Validation scripts automatically compare command outputs against expected results, with file changes verified via checksums. Preliminary experiments revealed that simpler tasks provided limited lifelong learning value; therefore, the final dataset focuses primarily on complex tasks to capture inter-skill dependencies.

**Knowledge Graph Environment:** This environment is based on a SPARQL query system. Tasks involve querying structured data through operations such as relation extraction and intersection. Tasks were curated from the GrailQA dataset (Gu et al., 2021) by mapping S-expressions to logical action sequences. These sequences range from 2 to 9 steps to ensure uniform distribution across task lengths. Each query is validated on a synthetic knowledge graph to confirm result correctness. Complex queries (7–9 steps) received additional manual validation to ensure semantic accuracy. Detailed description of each skill is provided in Appendix A.3.1.

### 4.3  QUALITY CONTROL

**Label Validation:** We employ automated validation mechanisms, including result comparison for SQL queries (Appendix A.1.2, Figure 5), exit code checking for Bash commands (Appendix A.2.2, Figure 6), and output verification for SPARQL queries (Appendix A.3.2). Additionally, 10% of tasks from each environment were manually reviewed for logical consistency and practical relevance. Pilot testing informed the final configuration to optimize task complexity and skill coverage. This multi-stage validation ensures that datasets are both challenging and representative of real-world scenarios.

**Balanced Skills:** In the Database environment, we generated 1,306 tasks with DeepSeek-R1 and selected 500 high-quality samples. Tasks cover 22 SQL skills with balanced distributions ensured by stratified sampling (Appendix A.1.2, Figure 7a). In the Operating System environment, 500 complex tasks were curated, with command sequences ranging from 9 to 12 steps to maximize inter-task skill overlap. Lower complexity tasks (1–8 steps) were excluded after preliminary tests showed minimal replay benefits (Figure 7b). In the Knowledge Graph environment, 396 tasks were extracted from GrailQA, mapped to atomic action sequences ranging from 2 to 9 steps. Replay effects were observed to diminish in sequences exceeding six steps.

## 5  EVALUATION FRAMEWORK

LifelongAgentBench is designed as a unified evaluation framework that integrates datasets and APIs to benchmark LLM-based agents under lifelong learning settings. In contrast to prior benchmarks that focus on static or single-task evaluation, our framework emphasizes sequential task execution and experience accumulation, offering a realistic simulation of continual learning scenarios. The detailed description is provided in Appendix B.

### 5.1  SYSTEM ARCHITECTURE

The framework comprises six loosely coupled components: **model pool** (Appendix B.1.1), **agent** (B.1.2), **environment** (B.1.3), **interaction history factory** (B.1.4), **controller** (B.1.5), and **callbacks** (B.1.6). Each component can be deployed independently across different servers and communicates via a custom remote procedure call (RPC) toolkit (B.2.1), enabling flexible distributed or local deployment (B.2.2).

The **model pool** component maintains mappings between model names and instances, supporting both open-source and proprietary LLM backends. The **agent** module translates environment observations and dialogue history into formatted inputs, queries the LLM, and parses outputs into executable actions. The **environment** component executes these actions and returns updated observations to the controller. It also implements standardized methods such as `reset`, `interact`, `complete`, `calculate_metric`, and `release` to ensure consistency across environments.

The **controller** component manages the interaction loop, oversees task scheduling, and relays agent actions to the environment. The **callback** component provides extensible hooks for monitoring internal events, facilitating reproducibility and experimental customization.

### 5.2  REPRODUCIBILITY AND MODULARITY

Two core design principles of LifelongAgentBench are **reproducibility** and **modularity**. The framework guarantees deterministic behavior under fixed random seeds and uses containerized environment snapshots to ensure identical task conditions across experimental runs. Additionally, it

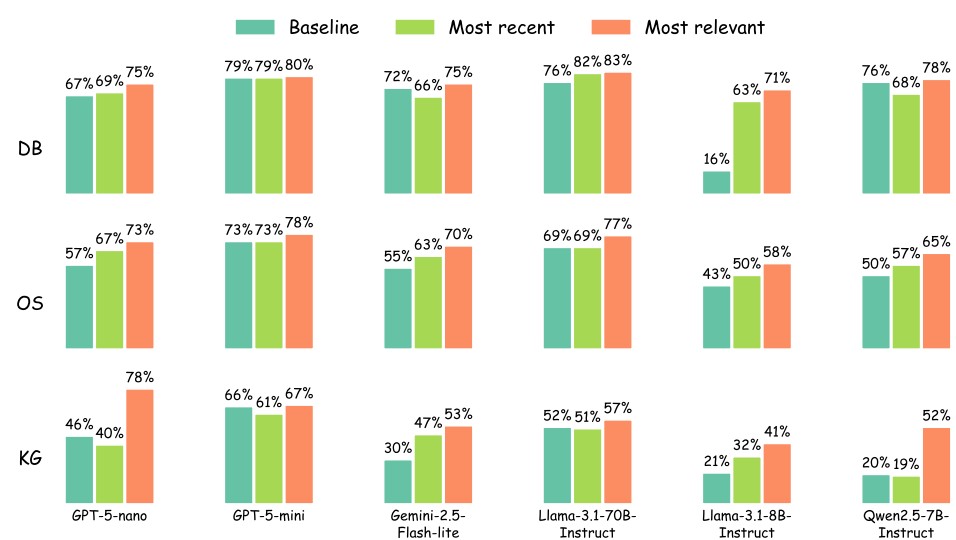

Figure 3: The comparison between different experience selection techniques. Selecting the most relevant experience yields higher performance than the most recent one.

exposes modular APIs for integrating new environments, custom agent architectures, or evaluation metrics with minimal engineering overhead. This flexibility allows researchers to experiment with various lifelong learning strategies while maintaining consistency and comparability.

### 5.3 Differences from Prior Benchmarks

Existing LLM agent benchmarks such as WebArena and AgentBench either rely on parallel task execution to reduce evaluation time or use process pools to manage multiple task sequences concurrently. These designs are incompatible with lifelong learning evaluation, where the strict order of task execution directly impacts the agent's accumulated knowledge and performance.

In contrast, LifelongAgentBench enforces strict sequential execution to preserve the integrity of experience accumulation and transfer learning assessments. Moreover, while prior frameworks tightly coupled agents, controllers, and environments into complex multi-process architectures, our design promotes developer-friendly single-process debugging with optional distributed scalability. This architecture substantially lowers the barrier to conducting lifelong learning research with LLM agents. The difference is summarized in Table 1.

## 6 Evaluation of LifelongAgentBench

We comprehensively evaluate the lifelong learning abilities of LLM-based agents using LifelongAgentBench across three environments: Database (DB), Operating System (OS), and Knowledge Graph (KG). Our experiments systematically investigate experience replay and group self-consistency under unified protocols.

### 6.1 Experimental Setup

**Models:** We evaluate the lifelong learning capabilities of a diverse set of LLM agents, encompassing both open-source and closed-source models as well as reasoning and non-reasoning variants. All agents share a unified API with reproducible initialization, dynamic experience replay, and optional group self-consistency. **Baselines and Metric:** We evaluate agents under baseline (no replay), experience replay (1, 4, 8, 16, 32, 64 prior trajectories), and experience replay with group self-consistency. Unless otherwise specified, the replayed experiences consist of the most recent successful trajectories. The evaluation metric is task success rate, defined as the proportion of correct action sequences that successfully complete the task. **Environments:** Experiments run on Linux

Table 2: The result of Llama-3.1-8B-Instruct on LifelongAgentBench. "Exp" represents that the number of recent **successful** trajectories that are provided to agent. The best result for each environment is bold. "OOM" represents out of memory.

|  | **Exp=0** | **Exp=1** | **Exp=4** | **Exp=16** | **Exp=32** | **Exp=64** |
|---|---|---|---|---|---|---|
| **DB** | 0.19 | 0.41 | 0.73 | 0.75 | 0.77 | **0.78** |
| **OS** | 0.43 | 0.46 | **0.50** | **0.50** | 0.42 | 0.44 |
| **KG** | 0.27 | **0.35** | 0.33 | OOM | OOM | OOM |

servers NVIDIA A800 (80GB). Code is based on Huggingface Transformers and PyTorch, with a distributed RPC-based framework for modular deployment (B.2.1). The system supports automatic checkpointing for recovery from interruptions.

## 6.2 MAIN RESULTS

**Evaluation of close-source and open-source models under baseline setting.** Figure 3 shows that the closed-source GPT-5-mini achieves the highest performance across all environments. Its advantage is particularly pronounced in the KG environment, where it outperforms the second-best model by 14%. Meanwhile, open-source models such as Llama-3.1-8B-Instruct also achieve reasonable and stable performance across all environments. This contrasts with prior benchmarks such as AgentBench, where open-source models often underperform, limiting academic reproducibility and accessibility.

**Experience replay consistently improves performance.** Table 2 shows that incorporating past successful trajectories consistently boosts agent performance compared to the baseline (Exp=0). For DB, replay increases accuracy from 19% to 78% at 64 examples. For OS, accuracy improves from 43% to 50% at 4–16 examples. For KG, accuracy improves from 28% to 35% with just 1 example.

**Trade-off between replay benefits and memory limitations.** Increasing replay beyond optimal values leads to diminishing or negative returns due to excessive input length, increased reasoning complexity, and out-of-memory (OOM) failures. DB tasks, which involve shorter trajectories, benefit from large replay buffers. In contrast, OS and KG tasks, with multi-turn and long-form interactions, show peak performance at lower replay sizes before degradation or OOM occurs.

**Memory-efficient replay remains an open challenge.** These findings suggest that while experience replay is a valuable mechanism for improving LLM agent performance, it introduces significant memory and inference costs. Designing more efficient retrieval and summarization strategies for lifelong learning remains an important avenue for future research.

## 6.3 EFFECT OF EXPERIENCE SELECTION TECHNIQUES

**Skills-based experience selection.** Results in Table 2 indicate that using the **most recent** successful trajectory as experience may introduce task-irrelevant information, which can confuse the agent. This suggests that selecting the **most relevant** trajectory may further enhance performance in lifelong learning settings. To test this hypothesis, we evaluate a broader set of agents using a simple skills-based selection strategy. For each task, we retrieve the past successful trajectory with the highest skill similarity (equation 1) to the current task. Because the optimal number of experiences varies across agents and environments, we assess each experience selection technique with 1, 4, and 16 experiences and report the best result for each. Due to the high invocation cost of closed-source models, we use only the first 100 tasks in each environment.

**Most relevant experience outperforms the most recent experience.** As shown in Figure 3, selecting the most relevant experience consistently leads to higher performance than selecting the most recent one. In some cases, providing the most recent experience even degrades performance relative to the baseline. In the DB and OS environments, the most recent experience offers noticeable improvements for models such as Llama-3.1-8B-Instruct, yet their performance can be further boosted by using the most relevant experience. The effect is most pronounced in the KG environment, where

Table 3: The result when using different backbone LLM. The environment is DB. The best result for each backbone LLM is bold. "OOM" represents out of memory.

| Model | Exp=0 | Exp=1 | Exp=4 | Exp=16 | Exp=32 | Exp=64 |
|---|---|---|---|---|---|---|
| DeepSeek-R1-Distill-Llama-8B | 0.07 | 0.13 | **0.35** | OOM | OOM | OOM |
| DeepSeek-R1-Distill-Qwen-7B | 0.10 | 0.12 | **0.18** | OOM | OOM | OOM |
| QwQ-32B | **0.83** | 0.80 | 0.80 | 0.81 | 0.79 | 0.82 |
| Qwen2.5-7B-Instruct | 0.74 | 0.71 | **0.76** | 0.74 | OOM | OOM |
| Qwen2.5-32B-Instruct | **0.82** | 0.77 | 0.71 | 0.72 | 0.74 | OOM |
| Llama-3.1-8B-Instruct | 0.19 | 0.41 | 0.73 | 0.75 | 0.77 | **0.78** |
| Llama-3.1-70B-Instruct | 0.81 | 0.83 | 0.86 | 0.88 | 0.88 | **0.90** |

models such as GPT-5-nano and Qwen2.5-7B-Instruct experience significant performance drops when given the most recent trajectory but achieve substantial gains when provided with the most relevant one. The experimental results further demonstrate that the **quality** of experience outweighs its **quantity**, highlighting the importance of developing more sophisticated experience selection strategies as a promising direction for future research on lifelong learning in LLM agents.

## 6.4 EFFECT OF MODEL BACKBONE

**Backbone LLMs.** We evaluate a range of open-source LLMs across different architectures and model scales. The results in Table 3 reveal notable differences in how backbone choice influences the effectiveness of experience replay. For strong base models such as Qwen2.5-7B-Instruct and Qwen2.5-32B-Instruct, the performance gain from adding prior experience is small or even negative. In particular, Qwen2.5-32B-Instruct achieves 0.82 accuracy without replay, with subsequent replay settings showing no clear improvement. In contrast, the Llama-3.1 series shows consistent and stable gains as more experience is provided, with Llama-3.1-70B-Instruct reaching 0.90 accuracy with 64 examples. This indicates that existing knowledge or architecture of backbone LLMs may substantially impact the utility of experience replay: some models may inherently learn well from single episodes, while others benefit more from historical trajectories.

**Model Size.** We observe a scaling trend where larger backbones consistently outperform their smaller counterparts across most replay settings. Llama-3.1-70B-Instruct demonstrates superior robustness, achieving the highest accuracy (0.90) without encountering OOM even at 64 replay examples. Interestingly, medium-scale models such as Llama-3.1-8B-Instruct achieve comparable performance (0.78 at 64 examples), suggesting that with careful experience replay and model tuning, smaller models can approach the performance of much larger ones while offering substantially lower computational cost.

## 6.5 SCALING EXPERIENCE WITH GROUP SELF-CONSISTENCY

**Group self-consistency reduces memory and stabilizes performance.** To mitigate the memory and inference overhead of large-scale experience replay, we propose group self-consistency, which splits retrieved experiences into smaller groups and aggregates their predictions using self-consistency voting (Wang et al., 2023). An illustration is provided in Figure 4. The experimental results are summarized in Tables 4.

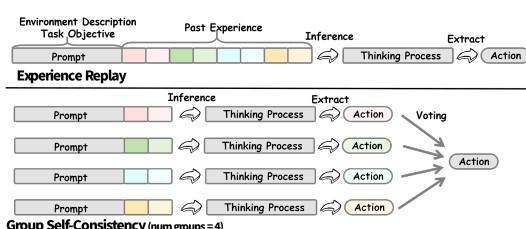

Figure 4: Illustration of group self-consistency.

**Significant accuracy gains in DB.** In DB, group self-consistency provides clear performance improvements as replay size increases. Llama-3.1-8B-Instruct achieves 0.75 accuracy with 16 groups (16 examples), compared to only 0.61 without grouping. Qwen2.5-7B-Instruct similarly benefits, improving from 0.72 to 0.77 accuracy. Smaller models such as DeepSeek-R1-Distill variants show limited gains, likely constrained by reduced model capacity.

Table 4: Comparison of the accuracy (average input tokens) under different group self-consistency settings.

| Environment | # Experience | 0 | 1 | 4 | | | 16 | | |
|---|---|---|---|---|---|---|---|---|---|
| | # Groups | / | 1 | 1 | 2 | 4 | 1 | 4 | 16 |
| DB | Llama-3.1-8B-Instruct | 0.16 (1128) | 0.51 (1773) | 0.63 (4189) | 0.57 (2750) | 0.59 (2512) | 0.61 (17874) | 0.70 (6008) | **0.75 (2888)** |
| | DeepSeek-R1-Distill-Llama-8B | 0.06 (1213) | 0.11 (2706) | 0.13 (6542) | 0.12 (4645) | 0.13 (3469) | 0.12 (27737) | 0.13 (10365) | **0.18 (4932)** |
| | Qwen2.5-7B-Instruct | 0.76 (951) | 0.69 (1567) | 0.68 (3952) | 0.73 (2445) | 0.73 (2292) | 0.72 (16383) | 0.71 (5443) | **0.77 (2685)** |
| | DeepSeek-R1-Distill-Qwen-7B | 0.11 (1232) | 0.09 (2353) | 0.16 (5830) | 0.26 (3861) | 0.12 (2292) | 0.15 (18845) | **0.23 (6113)** | 0.15 (2744) |
| KG | Llama-3.1-8B-Instruct | 0.27 (2978) | 0.32 (9390) | 0.26 (16420) | 0.33 (12175) | **0.36 (8842)** | 0.32 (56409) | 0.34 (22191) | 0.34 (11002) |
| | Qwen2.5-7B-Instruct | 0.22 (2858) | 0.27 (7150) | 0.19 (18436) | 0.29 (10998) | 0.30 (7522) | 0.19 (59950) | 0.19 (18539) | **0.36 (11339)** |

**Drastic memory savings in KG tasks.** In KG, where experience trajectories are much longer, group self-consistency dramatically reduces input token lengths. For Llama-3.1-8B-Instruct, token usage at 16 examples drops from 56,409 tokens (no grouping) to 11,002 tokens (16 groups), while maintaining stable accuracy. Qwen2.5-7B-Instruct shows similar trends, decreasing from 59,950 to 11,339 tokens with minimal accuracy loss.

**Group self-consistency offers a scalable replay strategy.** Overall, group self-consistency consistently stabilizes performance and alleviates memory bottlenecks across environments and models. Its simplicity and effectiveness make it a promising technique to enhance lifelong learning scalability for LLM agents under heavy replay loads. Future work may explore dynamic or adaptive grouping strategies that optimize the trade-off between experience diversity, inference cost, and available context window.

### 6.6 Additional experiments

We conduct additional experiments to examine the effects of task difficulty (Appendix E.1) and task order (Appendix E.2), as well as the transferability of experiences across environments (Appendix E.5). In addition, we compare the proposed group self-consistency method with vanilla self-consistency (Appendix E.4). We provide the additional results of o3-mini in Appendix E.3.

### 6.7 Failure Mode Analysis

To understand the failure patterns of LLM agents in LifelongAgentBench, we classify all task outcomes based on agent behavior and system status. We observe four common failure modes: (1) **Incorrect final submission**: the agent outputs an answer in the correct format but with wrong content (Appendix D.1). (2) **Failure to commit**: the agent completes multiple operations but never explicitly submits the final answer (Appendix D.2). (3) **Format violation**: the agent violates the required output format or instruction pattern (Appendix D.4). (4) **Context overflow**: the agent exceeds the LLM context window due to excessive interactions or large intermediate outputs (Appendix D.5). These results reveal key limitations of current LLM agents in multi-step interactive tasks: unstable reasoning, poor instruction adherence, and context management issues. Detailed examples of typical cases are provided in Appendix D.

## 7 Conclusion

We present LifelongAgentBench, the first unified benchmark specifically designed to evaluate the lifelong learning capabilities of LLM-based agents. Unlike prior benchmarks that treat agents as static systems, LifelongAgentBench systematically measures agents' ability to accumulate, retain, and transfer knowledge across diverse interactive environments. Our experiments demonstrate the potential of experience replay and group self-consistency to improve agent performance, while also revealing critical challenges.

## Ethics Statement

During the dataset construction process of LifelongAgentBench , although the closed-source LLMs used for data generation have undergone safety alignment and a portion of the constructed data was manually reviewed by the authors, it remains possible that the dataset contains content that could be

considered harmful or inconsistent with human ethical standards. We encourage users to exercise caution and apply appropriate filtering when using the dataset in future research.

## REPRODUCIBILITY STATEMENT

To promote reproducibility and encourage the reviewer process, all code, scripts, and the Life-longAgentBench dataset are provided in anonymous link `https://anonymous.4open.science/r/continual_agent_bench-4F3B` and will be made publicly available. We also provide a detailed description about the process of building datasets and experimental settings to promote reproducibility. This ensures that other researchers can validate our findings and build on our work, fostering collaboration and further advancements in the field.

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

APPENDIX

## A  LIFELONG LEARNING ENVIRONMENT DESIGN

In this section, we introduce the design principle and data construction process of each lifelong learning environment.

Table 5: Skill set in each environment.

| Environment | Skills |
|---|---|
| DB | column_alias, delete, group_by_multiple_columns, group_by_single_column, having_aggregate_calculation, having_multiple_conditions_with_aggregate, having_single_condition_with_aggregate, insert, limit_and_offset, limit_only, order_by_multiple_columns_different_directions, order_by_multiple_columns_same_direction, order_by_single_column, select, subquery_multiple, subquery_nested, ubquery_single, table_alias, update, where_multiple_conditions, where_nested_conditions, where_single_condition |
| OS | addgroup, awk, cat, cd, chage, chgrp, chmod, chown, chsh, cp, echo, exit, find, gpasswd, grep, groupadd, ln, ls, mkdir, mv, rm, sed, sleep, tee, touch, useradd, usermod, vi, wc |
| KG | get_relations, get_neighbors, intersection, get_attributes, argmax, argmax, count |

## A.1 DATABASE

### A.1.1 ENVIRONMENT INTRODUCTION

To ensure experiment reproducibility and reduce deployment complexity in the database environment, we encapsulate the MySQL database using Docker. At the start of each evaluation, the framework initializes a new Docker container from a predefined image. For each task, new tables are created based on task-specific information prior to execution and deleted upon task completion. This guarantees that modifications made during a task do not interfere with subsequent tasks. To optimize efficiency, the Docker container is launched only once at the beginning of the evaluation, rather than being recreated for each task, as the initialization of a MySQL container is time-consuming and frequent restarts would significantly increase overall evaluation time.

During each task, the agent interacts with the database based on the given goal $g$, performing operations such as querying, inserting, deleting, or updating records. For query tasks, the agent must submit the result at the end of the task, and the evaluation framework determines task success by comparing the submitted result with the ground truth. For other task types, the framework calculates a hash of the database state at the end of the task and compares it with a ground truth hash stored in the database to assess correctness.

In total, we define 22 skills within the database environment. The names and descriptions of these skills are as follows:

- **column_alias**: Assigning an alias to a column in a `SELECT` statement, e.g., `SELECT name AS employee_name`.
- **delete**: Deleting data using the `DELETE` statement.
- **group_by_multiple_columns**: Using the `GROUP BY` clause to group results by multiple columns.
- **group_by_single_column**: Using the `GROUP BY` clause to group results by a single column.
- **having_aggregate_calculation**: Including calculations on aggregate function results in the `HAVING` clause, e.g., `MAX(salary) - MIN(salary)`.
- **having_multiple_conditions_with_aggregate**: Including multiple conditions involving aggregate functions in the `HAVING` clause.
- **having_single_condition_with_aggregate**: Including a single condition based on an aggregate function in the `HAVING` clause.
- **insert**: Inserting data using the `INSERT` statement.
- **limit_and_offset**: Using both `LIMIT` and `OFFSET` clauses together.
- **limit_only**: Using only the `LIMIT` clause to restrict the number of returned rows, without using `OFFSET`.
- **order_by_multiple_columns_different_directions**: Sorting results by multiple columns with different sorting directions in the `ORDER BY` clause.
- **order_by_multiple_columns_same_direction**: Sorting results by multiple columns with the same sorting direction in the `ORDER BY` clause.
- **order_by_single_column**: Sorting results by a single column in the `ORDER BY` clause.

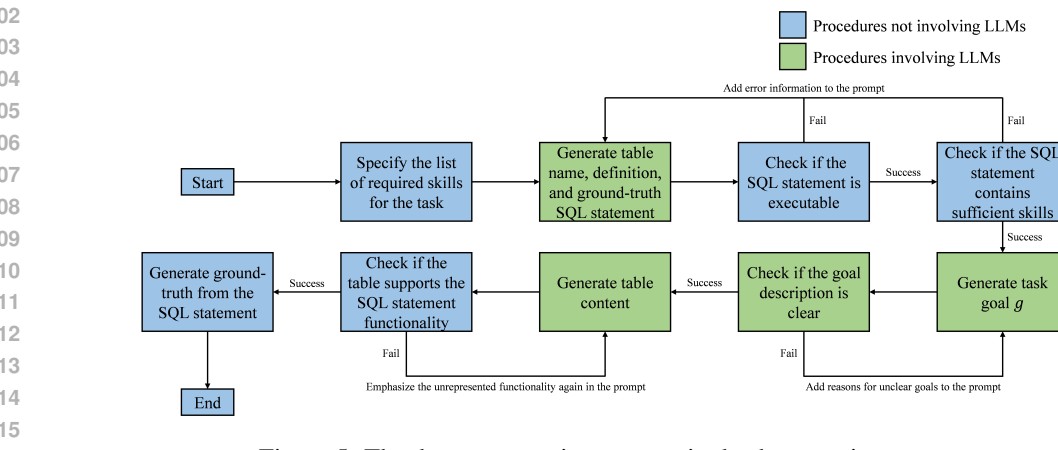

Figure 5: The data construction process in database environment.

- **select**: Querying data using the `SELECT` statement.
- **subquery_multiple**: Having multiple subqueries in the main query.
- **subquery_nested**: Nesting one subquery inside another subquery.
- **subquery_single**: Including a single subquery in the `SELECT`, `WHERE`, or `HAVING` clause that involves only one table.
- **table_alias**: Assigning an alias to a table in the `FROM` clause, e.g., `FROM employees AS e`.
- **update**: Updating data using the `UPDATE` statement.
- **where_multiple_conditions**: Including multiple conditions connected by `AND` or `OR` in the `WHERE` clause.
- **where_nested_conditions**: Including nested logical conditions in the `WHERE` clause, e.g., `WHERE (A AND B) OR C`.
- **where_single_condition**: Including a single condition in the `WHERE` clause, e.g., `WHERE id = 5`.

### A.1.2 DATA CONSTRUCTION PROCESS

The task construction process in the database environment is illustrated in Figure 5. To prevent data leakage, we adopt the methodology proposed in Tian et al. (2024), leveraging a large language model to generate the benchmark data.

During data construction, we begin by specifying a list of skills that each task should incorporate. To ensure balanced coverage, skills with lower occurrence frequencies are assigned higher sampling probabilities. Additionally, we enforce a minimum occurrence threshold for each skill across the dataset to avoid the presence of isolated tasks. The skill distribution is presented in Figure 7a.

### A.2 OPERATING SYSTEM

### A.2.1 ENVIRONMENT INTRODUCTION

In the operating system environment, we also use Docker containers to encapsulate the Ubuntu system with which the agent interacts. Due to the wide variety and broad impact of operations that the agent can perform in this environment, it is difficult to fully roll back the system state to its initial condition after each task. To address this, we destroy the current container instance after each task and instantiate a new one from the original image before the next task begins. This approach ensures both experimental reproducibility and isolation between tasks.

In the operating system environment, the agent is required to generate shell scripts composed of Bash commands based on the given goal $g$, in order to interact with the system. After each operation,

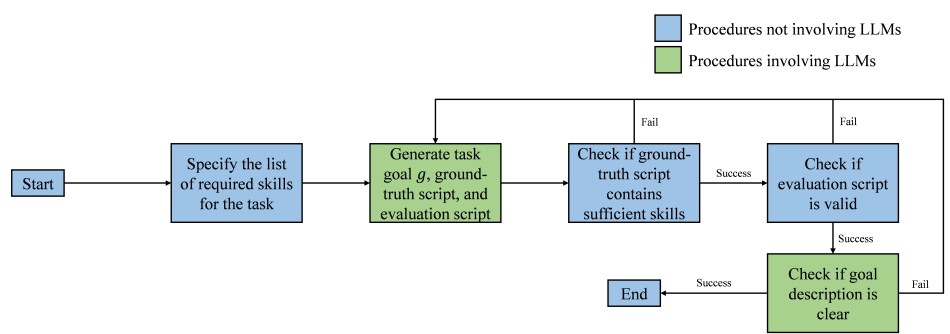

Figure 6: The data construction process in operating system environment.

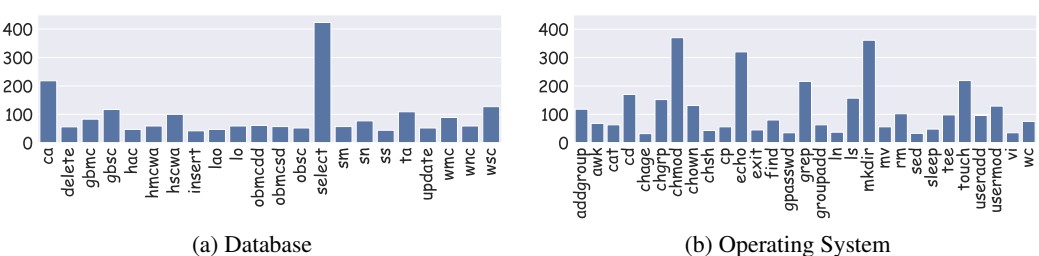

(a) Database             (b) Operating System

Figure 7: Skill distributions across tasks. Diverse and balanced skill coverage is maintained.

the agent can choose either to continue issuing shell commands or to terminate the interaction by outputting a string that matches a predefined pattern.

Similarly, if the number of interactions between the agent and the environment reaches a predefined limit, the evaluation framework automatically terminates the interaction. Regardless of the reason for termination, the framework then determines whether the agent has successfully completed the task. Each task is associated with an evaluation script consisting of Bash commands. If the agent completes the task successfully, the script returns 0; otherwise, it returns a non-zero value. The success of the task is thus determined based on the return value of the evaluation script.

In the operating system environment, each task is associated with a specific skill, determined based on the ground truth Bash command required to solve the task. To ensure controlled and consistent evaluation, we restrict the agent to a predefined set of 29 Bash commands during task execution, with each command corresponding to a distinct skill. The complete list of these 29 skills is presented in Table 5.

### A.2.2 DATA CONSTRUCTION PROCESS

The task construction process in the operating system environment is illustrated in Figure 6. As in the database environment, we use a large language model to generate the tasks in the dataset.

To reduce token costs, however, we minimize the number of calls to the language model during data construction in the operating system environment compared to the database environment. The skill distribution of the constructed dataset is presented in Figure 7a.

### A.3 KNOWLEDGE GRAPH

### A.3.1 ENVIRONMENT INTRODUCTION

In the knowledge graph environment, the agent is tasked with achieving the goal $g$ by interacting with the knowledge graph, given both the goal and the entities involved. While humans typically query a knowledge graph using SPARQL statements, generating a complete SPARQL query in a single step is relatively challenging for an agent. This difficulty arises from the agent's lack of

basic knowledge about the graph structure—such as entity types, outgoing relations, and incoming relations. Directly including all this information in the prompt results in excessively long inputs, which increases the risk of the agent overlooking critical details (Yang et al., 2025). Moreover, such a direct approach to knowledge graph access is impractical for real-world applications.

Following the approach proposed in Liu et al. (2023), we use a constrained set of actions that the agent can execute. This design enables the agent to progressively and autonomously explore relevant information within the knowledge graph and achieve the goal $g$ through a sequence of actions.

During the interaction between the agent and the knowledge graph environment, the environment returns sets of relations and entities based on the actions executed by the agent. Because the agent must reference previously retrieved relations to construct new actions in subsequent steps, all queried relations are returned in full. For entity sets, we use the concept of *variables* (Liu et al., 2023) to represent the results. Each variable is associated with the SPARQL statement that produced it, and instead of returning the set of queried entities directly, the environment returns the corresponding variable name to the agent. This mechanism allows the agent to conveniently reference previously queried entity sets in future actions, thereby enabling multi-hop reasoning over the knowledge graph.

In the knowledge graph environment, the set of available actions and their corresponding parameters is defined as follows. To complete a given task, the agent must execute a sequence of actions to achieve the goal $g$. Accordingly, we treat the actions appearing in the ground truth action sequence of each task as the task's skills.

- **get_relations**: Given an entity or variable, returns all outgoing relations connected to it.
- **get_neighbors**: Given an entity or variable and an outgoing relation, returns the set of entities reachable via that relation, represented as a new variable.
- **intersection**: Given two variables, returns a new variable representing the intersection of entities contained in both.
- **get_attributes**: Given a variable, returns all attributes associated with the entities in that variable.
- **argmax**: Given a variable and an attribute, returns a new variable containing the entity (or entities) with the maximum value for the specified attribute.
- **argmin**: Given a variable and an attribute, returns a new variable containing the entity (or entities) with the minimum value for the specified attribute.
- **count**: Given a variable, returns the number of entities it contains.

### A.3.2 DATA CONSTRUCTION PROCESS

In the knowledge graph environment, tasks generated solely by large language models often lack practical relevance. To address this, we construct tasks based on existing knowledge graph question answering datasets. Specifically, we adopt the dataset introduced in Gu et al. (2021) and convert the S-expressions representing graph queries into ground truth action sequences for each task.

## B EVALUATION FRAMEWORK DESIGN

After constructing the environment and tasks, it is essential to develop a robust evaluation framework to facilitate the interaction between LLM agents and the environment. In designing our framework, we drew inspiration from widely-used web-based agent benchmarks such as WebArena (Zhou et al., 2024), VisualWebArena (Koh et al., 2024b), as well as non-lifelong learning evaluation frameworks like AgentBench (Liu et al., 2023) and VisualAgentBench (Liu et al., 2024). However, these frameworks commonly suffer from several limitations, including long evaluation times, incompatibility with lifelong learning scenarios, and the lack of a unified development interface.

First, the evaluation time for a single sample is often prolonged. Take WebArena as an example. WebArena is a benchmark designed to assess the performance of LLM agents in real-world web environments, encapsulating each deployed website within a Docker container. However, due to the complexity of the webpages used, instantiating each Docker container takes approximately one minute. To mitigate this issue, WebArena claims to employ a carefully orchestrated task sequence

such that the execution of a preceding task does not affect subsequent ones. It therefore recommends initializing Docker containers only once before evaluation begins, rather than prior to each task. However, given that agent behaviors during evaluation are inherently unpredictable, determining whether tasks interfere with each other solely based on their goals $g$ is insufficient. Additionally, WebArena uses the accessibility tree of webpages as the environment observation. During the conversion to the accessibility tree, the evaluation framework traverses every HTML element in the DOM tree and maps it to a corresponding node. Even when using proprietary LLMs, the interaction time per task can reach up to ten minutes; for locally deployed open-source models, this duration increases further due to hardware constraints.

Second, these frameworks typically suggest evaluating agent performance via parallel execution of tasks, which is unsuitable for lifelong learning scenarios. For example, WebArena recommends partitioning the task sequence and running different sequences concurrently on separate servers. AgentBench manages all tasks using a central list and evaluates agent performance in parallel by distributing tasks across multiple processes in a process pool. When one process completes its assigned task, the framework automatically pulls the next one from the list. While these strategies reduce evaluation time, they fail to preserve task execution order—an essential factor in lifelong learning, where task sequencing determines the availability of transferable experience. Although both frameworks technically allow sequential task execution, doing so significantly increases overall evaluation time.

Third, these frameworks are primarily designed to assess the capabilities of different LLM agents, but do not provide a standardized interface for integrating or comparing various capability enhancement methods. This hampers fair comparisons. For instance, in WebArena, repeatedly resets webpage states within a single task using the built-in reset function to facilitate tree search. This practice is unrealistic, as real-world environments generally lack such reset mechanisms, and agent actions are often irreversible. Consequently, the comparison between the method (Koh et al., 2024a) and those proposed in other studies is inherently unfair. This stems from the framework's failure to provide a unified interface for the development of capability enhancement methods. Similarly, Agent-Bench separates the environment, controller, and agent into three independent processes that communicate via HTTP. Even if researchers have sufficient computational resources to deploy all components on a single machine, multi-process debugging introduces substantial development overhead. This complexity may explain why, compared to WebArena, fewer studies leverage AgentBench for investigating agent enhancement techniques.

To address these issues, we propose a new evaluation framework tailored to lifelong learning scenarios, aiming to support and accelerate the development of LLM-based agents and methods for continual capability improvement.

## B.1 FRAMEWORK ARCHITECTURE

The evaluation framework consists of six key components: the language model pool, the agent, the environment, the chat history item factory, the controller, and the callbacks. The callback module is further composed of a callback handler and a list of callback functions.

### B.1.1 LANGUAGE MODEL POOL

In the framework, the language model pool is implemented as a one-to-one mapping from string identifiers to language model objects. These model objects may belong to different language model classes, allowing for flexible integration of diverse models.

### B.1.2 AGENT

The agent component is responsible for converting the historical actions taken by the agent, as well as past observations received from the environment, into a textual input sequence. This sequence is then fed into the language model to generate the agent's next action through inference.

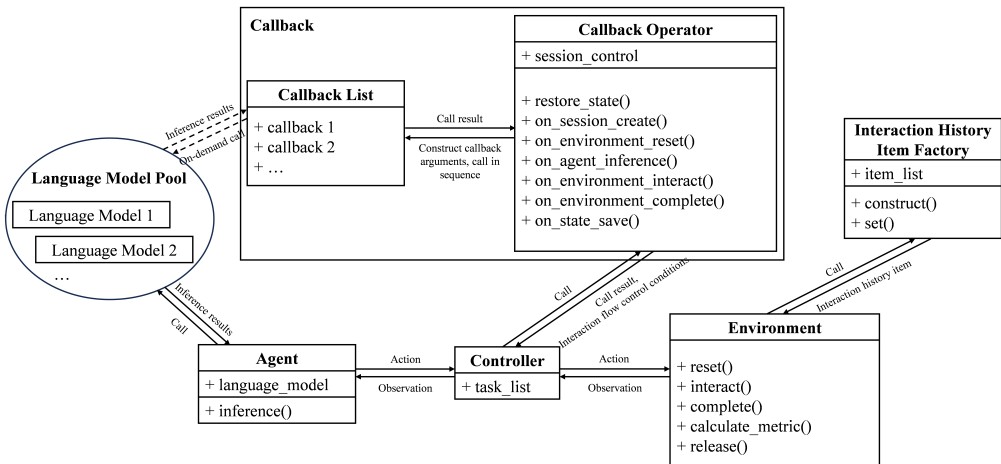

Figure 8: Evaluation Framework Architecture

### B.1.3 ENVIRONMENT

The environment component is responsible for executing the agent's actions and returning the corresponding responses to the controller. It provides five core methods: `reset`, `interact`, `complete`, `calculate_metric`, and `release`. Specifically:

- **reset**: Initializes the environment to the initial state $s_0$ at the beginning of each task and returns the task goal $g$. In environments that utilize Docker containers (e.g., the operating system environment), this method also creates a new container for agent interaction.

- **interact**: Executes the agent's action and returns the resulting observation from the environment to the controller.

- **complete**: Evaluates whether the agent has successfully completed the task based on the goal-task reward function $R$. In environments where a new Docker container is created for each task (e.g., the operating system environment), this method also handles the destruction of the container. In environments that reuse a single container across tasks (e.g., the database environment), it restores the container to its initial state.

- **calculate_metric**: Assesses the agent's overall performance within a lifelong learning scenario, based on the cumulative rewards obtained across all tasks.

- **release**: Used in environments that do not require per-task container creation. It performs a unified cleanup by destroying the container to free system resources.

All test environments introduced in Section 4.2 share a common abstract parent class. Each environment must implement all abstract methods defined by this parent class and may only expose the public methods specified therein to other components. This design not only facilitates the extension of the evaluation framework to support new environments, but also encourages the development of methods for enhancing the capabilities of large language model-based agents that are broadly applicable across various lifelong learning environments.

### B.1.4 INTERACTION HISTORY ITEM FACTORY

The interaction history item factory component is responsible for initializing the interaction history based on the environment description and task goal $g$. During the environment's reset process, the construct method of the factory is called to generate the initial interaction history by combining the task goal $g$ with the initial observation $O_0$

Recent research has shown that incorporating historical experience into prompts or dynamically selecting prompt strategies based on environmental observations can significantly enhance the performance of LLM-based agents. To promote modular system design and decouple the logic of

interaction history construction from environment-specific logic, we abstract this functionality into an independent interaction history item factory component. This design improves both the clarity and extensibility of the framework's codebase.

### B.1.5  CONTROLLER

The controller component is primarily responsible for creating sessions, executing tasks, and storing the execution history of previously completed tasks.

A session is a structured object that encapsulates a task. In addition to recording key information such as the task index, interaction history, and the reward obtained by the agent, each session also maintains the session state that indicates its current status and the reason for task termination. As the task progresses, the interaction history grows incrementally. Both the agent and environment components can update the interaction history to exchange newly generated actions and observations, and they can also set the session state to notify the controller of any exceptional conditions encountered during task execution. Upon completion, each session is saved to a history list for future reference.

LLM-based experiments often span several hours or even days. To address this, our framework incorporates a snapshot mechanism that periodically persists critical data to disk at predefined intervals. This allows for efficient recovery and resumption in the event of unexpected interruptions or failures. Upon experiment startup, the controller attempts to restore any previously interrupted experiment. If the experiment is recoverable, the controller reads the necessary data from disk and skips already completed tasks, continuing with the remaining ones. After each task is completed and its corresponding session is added to the history list, the controller immediately saves the session data to disk. With this mechanism, an unexpected interruption results in the loss of, at most, a single task's interaction data.

The controller is also responsible for calling callbacks at pre-defined events during task execution. We detail the callback mechanism in Appendix B.1.6.

### B.1.6  CALLBACK

Inspired by the `Trainer` module in the *transformers* library, we designed a callback component for the evaluation framework. This component can manage multiple callbacks, each of which can modularly implement a specific function.

We manage all callbacks through a centralized list. Key points in task execution are defined as *events*, and when an event occurs, all callbacks in the list are invoked sequentially. Each callback inherits from a common parent class, which provides event handler functions named after each event. By default, these handler functions do not affect task execution. However, researchers can inherit from the base callback class and override the event functions of interest. The customized callbacks can then be added to the callback list to influence task behavior and implement methods for enhancing agent capabilities.

This design ensures that all lifelong learning methods are developed and evaluated under consistent assumptions, thereby promoting comparability and fairness across experiments. Furthermore, since callbacks in the list are isolated and unaware of each other, researchers can combine multiple callbacks from different methods within a single experiment to explore the synergistic effects of various approaches on agent performance.

Each event handler function receives five arguments: the agent, the environment, the current session, the session history list, and the session controller. The agent and environment refer to the components described in Appendix B.1.2 and Appendix B.1.3. The current session encapsulates a task, as detailed in Appendix B.1.5. The history session list is a key parameter for lifelong learning. Event handler functions are expected to extract useful experience from this list and enhance the large language model-based agent's performance in lifelong learning scenarios by modifying the interaction history of the current task.

The session controller, a data member accessible to callbacks, consists of a set of boolean flags that can be used to skip specific steps in the interaction between the agent and the environment. For example, if an event handler detects that the agent's response does not contain a valid action that the

environment can interpret, it can use the session controller to skip the step of sending the action to the environment and instead prompt the agent to regenerate its response.

Notably, our framework allows each callback to optionally hold a reference to the language model during initialization and call it when needed. In contrast, most existing LLM-agent evaluation frameworks (e.g., AgentBench) do not decouple the LLM from the agent; instead, they treat the LLM-based agent as a single monolithic component. This design choice hinders the development of methods aimed at enhancing agent capabilities. For example, if a single LLM $L_1$ is used by an agent and two different methods, each requiring LLM $L_2$ access, are applied simultaneously via separate callbacks, and no model pool exists, both methods would redundantly load identical model arguments into GPU memory, as the callbacks are unaware of each other.

Additionally, if a researcher wants to perform multiple forward passes within a callback to improve response quality, calling the agent's inference API directly may be inappropriate, since it typically only supports inference for a single dialogue history, not for batched parallel queries.

With the introduction of a shared language model pool, identical language model parameters used by both callbacks and the agent are loaded into GPU memory only once. This significantly reduces hardware requirements for developing agent capability enhancement methods and supports better modularity in the framework design.

We define a total of seven events for the callback mechanism. These events are as follows. During a single run of the evaluation framework, the *restore_state* event occurs at most once. Within the execution of a single task, *on_session_create*, *on_environment_reset*, *on_environment_interact*, and *on_state_save* each occur once, In contrast, *on_agent_inference* and *on_environment_interact* may be triggered multiple times throughout the task.

- `restore_state`: During the execution of an experiment, not only the controller's state but also the internal states of the callbacks (i.e., the values of their data members) may change. Callbacks should save their internal state to disk when the `on_state_save` event occurs and reload it when the `restore_state` event is triggered.
- `on_session_create`: This event is triggered after a session is initialized. At this point, the session only contains the index of the assigned task.
- `on_environment_reset`: This event is triggered after the `reset` method of the environment is called.
- `on_agent_inference`: This event is triggered after the agent's `inference` method is called.
- `on_environment_interact`: This event is triggered after the environment's `interact` method is called.
- `on_environment_complete`: This event is triggered after the environment's `complete` method is called.
- `on_state_save`: This event occurs when the controller writes its state to disk.

## B.2 DIFFERENCES BETWEEN DISTRIBUTED DEPLOYMENT AND SINGLE-MACHINE DEPLOYMENT

Nearly all LLM-based agent evaluation frameworks, including the one we propose, encapsulate the interactive environments for agents using Docker containers. However, due to the large number of parameters in LLM agents, they typically need to be deployed on high-performance computing (HPC) nodes. Most HPC nodes do not allow users to run Docker applications. As a result, most existing evaluation frameworks for LLM-based agents allow researchers to deploy the environment and the LLM on separate servers. WebArena (Zhou et al., 2024) and VisualWebArena (Koh et al., 2024b) , which only consider environments in the form of webpages, deploy Docker containers containing the webpages on external servers, and then use headless browsers running on the LLM-hosting server to access those webpages. However, this approach clearly cannot be generalized to other types of environments. AgentBench (Liu et al., 2023) and VisualAgentBench (Liu et al., 2024) support distributed deployment via custom-built Remote Procedure Call (RPC) toolkit. Although their RPC toolkit can handle diverse environments such as web pages and databases, the introduction of RPC brings additional cognitive load to developers. Even when researchers use servers capable of

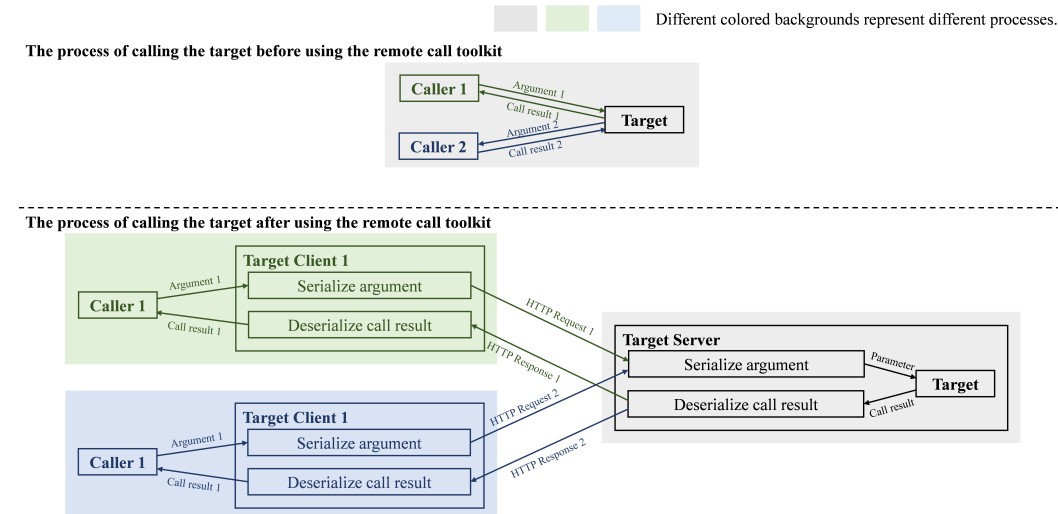

Figure 9: The Implementation of our RPC toolkit

running Docker and opt for small-parameter LLMs to develop agent enhancement methods, the use of RPC forces them to run different components of the evaluation framework in separate processes, significantly complicating debugging.

To address these issues, our evaluation framework is designed to support both single-machine and distributed deployment modes. During development, researchers can deploy all components on the same server and within a single process, enabling rapid prototyping and easier debugging. During evaluation, however, the framework also allows different components to be deployed across multiple servers and processes, making it possible to evaluate agents powered by large-scale language models in lifelong learning scenarios.

Crucially, the use of RPC should be transparent to researchers during development. The transition from local to distributed deployment should require only the execution of a small amount of additional, highly reusable code, allowing the framework to switch deployment modes seamlessly without introducing unnecessary complexity.

To achieve these goals, we designed and implemented a plug-and-play RPC toolkit based on a client-server architecture. During development, researchers can choose not to use this toolkit and debug all components of the framework within the same process. They can also use the toolkit to split components that need to be deployed on other servers into a client and a server, and convert component calls into remote procedure calls to enable distributed deployment of the evaluation framework. Throughout the development process, researchers can operate under the assumption that all components run within the same process, without needing to understand the internal details of the toolkit or RPC mechanisms. The plug-and-play nature of the toolkit allows effortless switching between single-machine and distributed deployments. The plug-and-play RPC toolkit helps reduce the cognitive load on researchers during development, shortens debugging time, and promotes the proposal of new agent enhancement methods in lifelong learning scenarios.

We first introduce the RPC toolkit we implemented, and then introduce the structure of the framework under distributed deployment.

### B.2.1 REMOTE PROCEDURE CALL TOOLKIT

The implementation of RPC toolkit is shown in the Figure 9. In the figure, "client," "server," and "target" all refer to instantiated objects, but for brevity, we omit the word "object." We refer to the class that needs to be remotely called as the "target class." The target class can provide methods for other components of the evaluation framework to call. For each object of the target class (i.e., the target object), we wrap it with an object of the target server class (i.e., the target server object). Callers within the evaluation framework can access the target through the corresponding target client

class object (i.e., the target client object). All target server classes and client classes are subclasses of the server and client base classes provided by our RPC toolkit. To implement RPC functionality, each target class must define its corresponding target server subclass and client subclass.

The target server continuously listens on a specific port on the server. When a caller calls a method of the target via the target client, the client first serializes the arguments and sends them via HTTP to the port that the target server is listening on. Upon receiving the HTTP request, the server deserializes the arguments into a format that the target can directly use and then performs the method call. After the call is completed, the result is returned to the target server, which then serializes the result and returns it via HTTP to the target client. The client then deserializes the result to obtain the correctly formatted output. The serialization and deserialization processes are mainly handled by the server and client base classes. The toolkit also supports reading and modifying data members in the same manner.

When implementing the server and client subclasses for a target, researchers only need to ensure that the target client provides the same interface as the target and defines the parameter types and return types of each interface using the `BaseModel` class from the *Pydantic* library. This makes it possible to implement remote invocation of the target without needing to understand the specifics of the serialization and deserialization processes. Through this design, we decouple the process of data transfer between processes during remote invocation from the target's data processing logic.

The RPC toolkit supports serialization and deserialization of various object types, including immutable built-in Python types, enumeration types, and any subclass of the client base class. We implement the client base class as a subclass of the `BaseModel`. Consequently, all concrete client classes are also subclasses of `BaseModel`. The `BaseModel` class requires developers to explicitly specify the types of data members via type annotations and automatically performs type validation during object construction. This ensures that objects maintain data integrity and type consistency after serialization and deserialization.

By supporting serialization and deserialization of client subclass objects, our toolkit allow client subclass objects to be used as return values in remote procedure calls and passed back to the caller. This enables researchers to conveniently implement chained remote procedure calls, where the result obtained from one remote call can be directly used to initiate another remote procedure call, without the need to explicitly instantiate the client subclass object during development. The instantiation process is handled automatically by the toolkit, further simplifying the implementation.

The process of chained remote calls is illustrated in the Figure 10. In the following example, there are two target objects, $P_a$ and $P_b$. $P_a$ contains $P_b$ as a data member. $P_a$ and $P_b$ can be two objects of the same target class or of different target classes. We assume the caller *caller* wants to call the *method* of $P_b$ via $P_a$, and *caller*, $P_a$, and $P_b$ are located in three different processes. Through the toolkit, what *caller* actually accesses is the client object $C_a$ corresponding to $P_a$, and what $P_a$ actually stores is the client object $C_b$ corresponding to $P_b$.

Note that during development, researchers can still think of *caller* as accessing $P_a$ rather than $C_a$, and $P_a$ as containing $P_b$ as a data member rather than $C_b$. When using the dot operator and calling $P_b$'s *method*, a remote procedure call is triggered. The dot operator takes a name string as input and returns the corresponding data member. In Figure 10, the server objects corresponding to $P_a$ and $P_b$ are denoted as $S_a$ and $S_b$ respectively, and $C_b'$ is the internally instantiated client of $P_b$ created by the toolkit, which is transparent to the researcher and used to implement the chained remote call.

In the above remote procedure call process, all steps except for Step 3, Step 7, and Step 10 are implemented by the toolkit and are transparent to the researchers.

### B.2.2 DISTRIBUTED DEPLOYMENT FRAMEWORK STRUCTURE

The architecture of the evaluation framework under distributed deployment is illustrated in the Figure 11. For clarity, some internal details of the components and the RPC toolkit are omitted. During experiments, the client-side controller process typically runs on a high-performance computing node, while the controller server, interaction history item factory server, and environment server processes usually run on standard servers with Docker support. Communication between these distributed processes is handled by the RPC toolkit described in Appendix B.2.1.

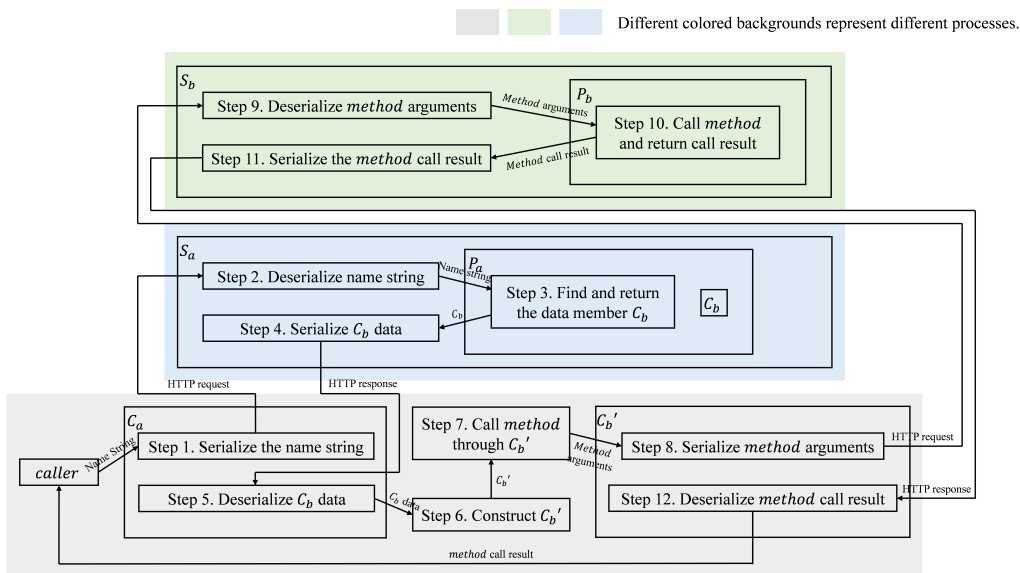

Figure 10: Example of Using our RPC Toolkit

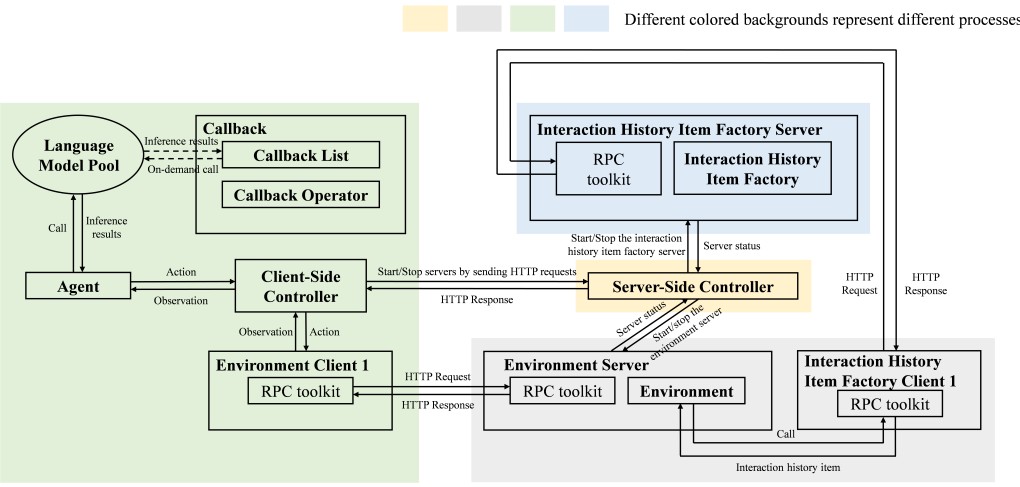

Figure 11: The Architecture of the Evaluation Framework in a Distributed Deployment

Before the experiment begins, researchers first need to launch the server-side controller on a standard server, followed by launching the client-side controller on the high-performance computing node. The server-side controller continuously listens on a designated port. When the client-side controller starts, it sends an HTTP request to the server-side controller to start the environment server and the interaction history item factory server. Once the client-side controller receives confirmation of successful server initialization, it begins task scheduling. After all tasks are scheduled, the client-side controller sends another HTTP request to the server-side controller to shut down the interaction history item factory server and the environment server.

The end of an experiment corresponds to the termination of the client-side controller, interaction history item factory server, and environment server processes. In contrast, the server-side controller process continues to run. This design allows the server-side controller to continuously listen on a designated port and, upon the next launch of the client-side controller, reinitialize the environment server and interaction history item factory server based on the configuration file provided. In the distributed deployment, the client-side controller can be viewed as a wrapper around the single-

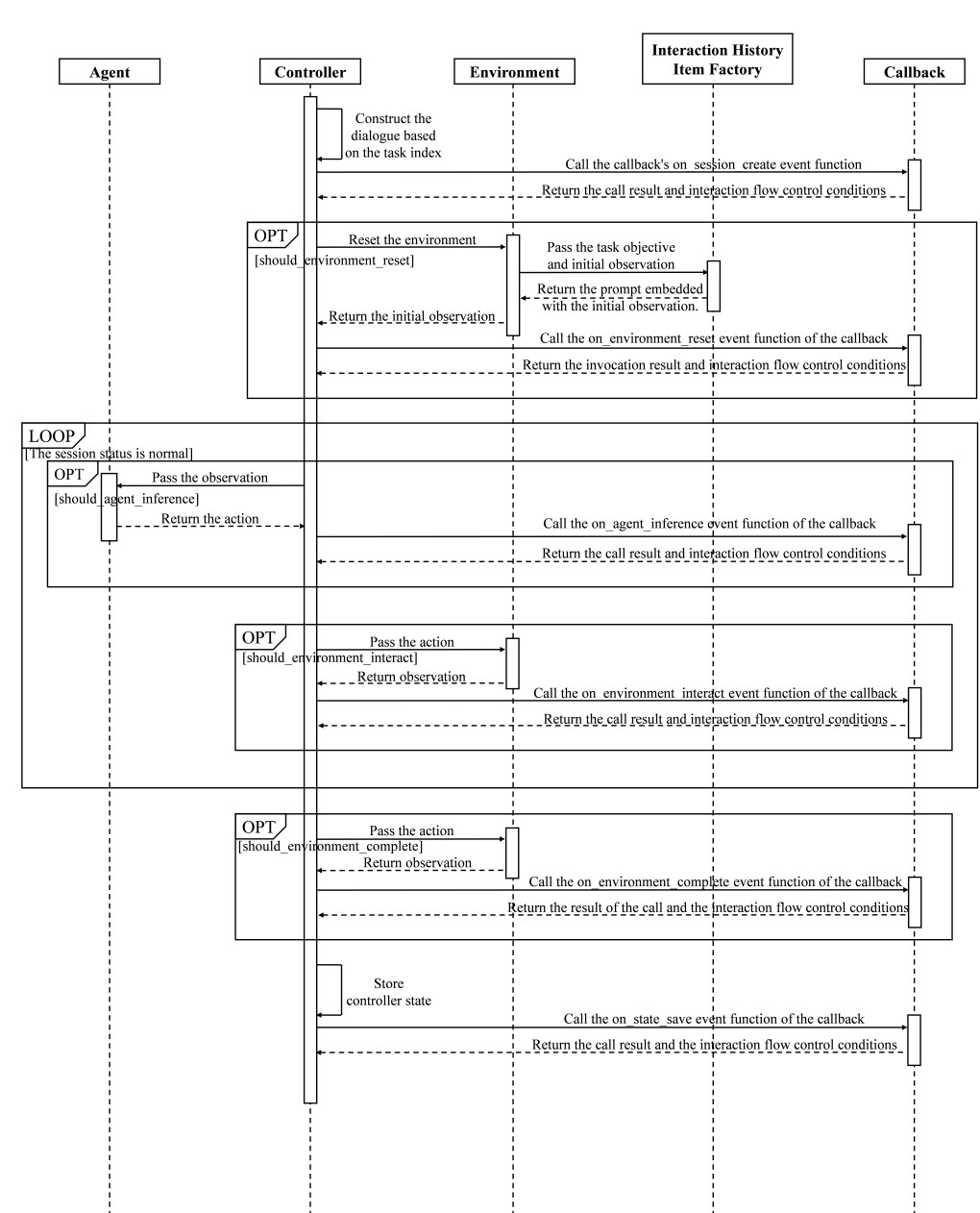

Figure 12: The execution process of an assessment task

machine controller, with additional functionality for starting and shutting down the necessary servers at the beginning and end of each experiment.

During the implementation of the evaluation framework, we found that if the interaction history item factory component is treated as a regular data member of the environment component and not wrapped in a server subclass from our RPC toolkit, modifications made to the interaction history item factory on the client side will not be synchronized with the environment server. Specifically, without running the interaction history item factory component in a separate process, the client-side controller's process can easily obtain a copy of the interaction history item factory that is identical to the one in the environment component. However, this copy is merely a local mutable variable on the frame of the client-side controller process, and any modifications made to its data members will

not be synchronized with the interaction history item factory in the environment server. To resolve this issue, we run the interaction history item factory component in a separate process. Callers in the process where the environment server resides can access and modify the interaction history item factory through remote procedure calls. Meanwhile, callers in the process where the client-side controller resides can access and modify the interaction history item factory via chained remote procedure calls.

### B.3 THE EXECUTION PROCESS OF A TASK

To clearly demonstrate the execution process of the evaluation framework and show the specific timing of events during task execution, we present the execution process of a task in the evaluation process in Figure 12. We assume that the evaluation framework is deployed on a single machine, so the components interact directly with each other without the need for the server or client in the RPC toolkit. In distributed deployment, for the researchers, the interaction methods and arguments passed between components remain the same as in a single-machine deployment.

In the evaluation framework, the parameters of the callback component are passed by reference, so the callback does not explicitly return results to the controller. In the diagram, `should_environment_reset`, `should_agent_inference`, `should_environment_interact`, and `should_environment_complete` are the interaction flow control conditions determined by the callback component. All four interaction flow control conditions default to true. Callbacks can modify these control flags to influence the interaction process between the agent and the environment.

## C DATASETS SAMPLES

### C.1 DATABASE SAMPLES

---

**Environment: DB —— Case 1**

Instruction: Insert a new payment record for member ID 102 with a payment date of "2023-10-15", amount 75, and payment method "Credit Card" into the membership payments table. \nThe name of this table is membership_payments, and the headers of this table are payment_id, member_id, payment_date, amount, payment_method.

```
Ground Truth Action:  INSERT INTO membership_payments
(member_id, payment_date, amount, payment_method) VALUES
(102, "2023-10-15", 75, "Credit Card");
```

Skill List: ["insert"]

---

**Environment: DB —— Case 2**

Instruction: Delete all entries where maintenance_status is equal to 0.\nThe name of this table is equipment_service, and the headers of this table are equipment_id, equipment_type, installation_date, maintenance_status, service_interval.

```
Ground Truth Action:  DELETE FROM equipment_service WHERE
maintenance_status = 0;
```

Skill List: ["delete", "where_single_condition"]

---

**Environment: DB —— Case 3**

Instruction: Update the result status to "completed" and hours spent to 48 for the experiment with ID equal to 105 in the experiment_data table.\nThe name of this table is experiment_data, and the headers of this table are experiment_id, researcher_name, experiment_date, result_status, hours_spent, samples_used.

```
Ground Truth Action:  UPDATE experiment_data SET
result_status = "completed", hours_spent = 48 WHERE
experiment_id = 105;
```

Skill List: ["update", "where_single_condition"]

---

Environment: DB —— Case 4

Instruction: What are the models, years, and prices of vehicles that are currently available? Return the model, year, and price, ordered by price from highest to lowest.\nThe name of this table is vehicle_inventory, and the headers of this table are vehicle_id, model, year, price, status.

```
Ground Truth Action:  SELECT model, year, price FROM
vehicle_inventory WHERE status = "available" ORDER BY price
DESC;
```

Skill List: ["order_by_single_column", "select", "where_single_condition"]

---

Environment: DB —— Case 5

Instruction: What are the property listings' id (listing ID), type (property type), price, and square feet? Return the results ordered by price from highest to lowest, then square feet from highest to lowest, and limit the output to 10 entries.\nThe name of this table is property_listings, and the headers of this table are listing_id, property_type, price, square_feet, listed_date, status, agent_id.

```
Ground Truth Action:  SELECT listing_id AS id, property_type
AS type, price, square_feet FROM property_listings ORDER BY
price DESC, square_feet DESC LIMIT 10;
```

Skill List: ["column_alias", "limit_only", "order_by_multiple_columns_same_direction", "select"]

---

Environment: DB —— Case 6

Instruction: Which routes have a distance greater than the average distance and less than the maximum distance? Return the route ID and driver name.\nThe name of this table is delivery_routes, and the headers of this table are route_id, driver_name, vehicle_type, distance_km, delivery_status.

```
Ground Truth Action:  SELECT route_id, driver_name
FROM delivery_routes WHERE distance_km > (SELECT
AVG(distance_km) FROM delivery_routes) AND distance_km <
(SELECT MAX(distance_km) FROM delivery_routes);
```

Skill List: ["select", "subquery_multiple", "where_multiple_conditions"]

---

Environment: DB —— Case 7

Instruction: What are the owner IDs and the total number of vaccinations administered to dogs since January 1, 2023? Return the owner ID and corresponding total vaccinations, only including owners with more than 3 vaccinations.\nThe name of this table is vaccination_records, and the headers of this table are record_id, owner_id, pet_type, vaccination_type, vaccination_date, veterinarian, dose_number.

```
Ground Truth Action:  SELECT owner_id, COUNT(*) AS
total_vaccinations FROM vaccination_records WHERE pet_type =
```

```
"Dog" AND vaccination_date >= "2023-01-01" GROUP BY owner_id
HAVING COUNT(*) > 3;
```

Skill List: ["column_alias", "group_by_single_column", "where_multiple_conditions", "having_single_condition_with_aggregate", "select"]

---

**Environment: DB —— Case 8**

Instruction: What are the item IDs and names for items that are either: 1) in the "Electronics" category with stock quantity below the average stock quantity of all items, or 2) in the same category as the item with ID 100 (determined dynamically via its category)? Return the item ID and name.\nThe name of this table is inventory_management, and the headers of this table are item_id, item_name, category, stock_quantity, last_restock_date.

```
Ground Truth Action:  SELECT i.item_id AS id, i.item_name
AS name FROM inventory_management AS i WHERE (i.category
= "Electronics" AND i.stock_quantity < (SELECT
AVG(stock_quantity) FROM inventory_management)) OR i.category
= (SELECT category FROM inventory_management WHERE item_id =
100);
```

Skill List: ["column_alias", "subquery_multiple", "table_alias", "where_nested_conditions", "select"]

---

**Environment: DB —— Case 9**

Instruction: Which seat numbers have a price difference (maximum price minus minimum price) greater than 50? Return the seat number and price difference, ordered by seat number in descending order, and limit the results to 5 entries.\nThe name of this table is ticket_sales, and the headers of this table are ticket_id, seat_number, price, sale_date, customer_name, status.

```
Ground Truth Action:  SELECT seat_number,
MAX(price)-MIN(price) AS price_diff FROM ticket_sales GROUP
BY seat_number HAVING MAX(price)-MIN(price)>50 ORDER BY
seat_number DESC LIMIT 5;
```

Skill List: ["column_alias", "group_by_single_column", "having_aggregate_calculation", "having_single_condition_with_aggregate", "limit_only", "order_by_single_column", "select"]

---

**Environment: DB —— Case 10**

Instruction: What are the guest IDs, their average ratings, and total reviews for guests who have either given a 5-star rating or submitted a review before January 1, 2023, and have more than 5 total reviews with an average rating below 4.5? Return the results ordered by average rating in descending order and total reviews in ascending order.\nThe name of this table is hotel_reviews, and the headers of this table are review_id, guest_id, review_text, rating, review_date.

```
Ground Truth Action:  SELECT guest_id AS gid, AVG(rating)
AS avg_rating, COUNT(review_id) AS total_reviews FROM
hotel_reviews AS hr WHERE guest_id IN (SELECT guest_id
FROM hotel_reviews WHERE rating = 5) OR guest_id IN (SELECT
guest_id FROM hotel_reviews WHERE review_date < "2023-01-01")
GROUP BY guest_id HAVING COUNT(review_id) > 5 AND AVG(rating)
< 4.5 ORDER BY avg_rating DESC, total_reviews ASC;
```

Skill List: ["where_multiple_conditions" "group_by_single_column", "select", "table_alias", "subquery_multiple", "column_alias", "having_multiple_conditions_with_aggregate", "order_by_multiple_columns_different_directions"]

## C.2 Operating System Samples

**Environment: OS —— Case 11**

Instruction: Create three files "/tmp/file1", "/tmp/file2", and "/tmp/file3", setting each file's permissions to 600 with a 1-second delay after creating each file. Use the "sleep" command between file operations.

Ground Truth Action: `rm -f /tmp/file1 /tmp/file2 /tmp/file3`

Skill List: ["chmod", "sleep", "touch"]

**Environment: OS —— Case 12**

Instruction: Recursively change ownership of "/var/webapp" to user "webadmin", set directory permissions to 750, and file permissions to 640. Ensure hidden files in subdirectories are included.

```
Ground Truth Action:  mkdir -p
/var/webapp/{public,private/.cache} &&
touch /var/webapp/public/index.html
/var/webapp/private/.cache/temp.dat &&
echo "config" > /var/webapp/private/.env
```

Skill List: ["chown", "find", "useradd"]

**Environment: OS —— Case 13**

Instruction: Modify the Nginx configuration file at /etc/nginx/sites-available/default with the following changes using sed: 1) Replace all "http://" with "https://", 2) Change the listen port from 80 to 8080, 3) Set server_name to "myapp.com", 4) Disable server_tokens by setting it to "off", 5) Add "client_max_body_size 20M;" after the server block opening, 6) Insert "keepalive_timeout 65;" before the location block, 7) Update the proxy_pass directive to use HTTPS, 8) Add a comment "# Security update" before server_name, and 9) Ensure all changes are made in-place. Preserve the original configuration structure.

```
Ground Truth Action:  mkdir -p /etc/nginx/sites-available
&& printf 'server {\n listen 80;\n server_name
example.com;\n server_tokens on;\n location /
{\n proxy_pass http://backend;\n }\n }\n ' >
/etc/nginx/sites-available/default
```

Skill List: ["cd", "cp", "echo", "sed"]

**Environment: OS —— Case 14**

Instruction: 1. Create a group "testgroup", add "testuser" to it, set group ownership recursively on "/var/www/project" to "testgroup", set directory permissions to 775, file permissions to 660, and enable setgid on the project directory.

```
Ground Truth Action:  useradd -m testuser
&& mkdir -p /var/www/project/{assets,logs}
&& touch /var/www/project/assets/image1.jpg
```

```
/var/www/project/logs/debug.log && chmod 755 /var/www/project
```

Skill List: ["addgroup", "chgrp", "chmod", "find", "usermod"]

---

**Environment: OS —— Case 15**

Instruction: Remove /tmp/cleanup/file1.tmp and /tmp/cleanup/dir1/file2.tmp, delete all empty directories within /tmp/cleanup, and ensure the /tmp/cleanup directory exists.

```
Ground Truth Action: mkdir -p /tmp/cleanup/dir1
/tmp/cleanup/dir2 && touch /tmp/cleanup/file1.tmp
/tmp/cleanup/dir1/file2.tmp /tmp/cleanup/dir2/file3.log
```

Skill List: ["echo", "exit", "find", "ls", "mkdir", "rm"]

---

**Environment: OS —— Case 16**

Instruction: Create a group "devteam", add users "user1", "user2", and "user3" to it, create a directory "/var/devteam_projects" accessible only to the group, and generate a log file with group details.

```
Ground Truth Action: useradd -m user1 && useradd -m user2 &&
useradd -m user3
```

Skill List: ["addgroup", "chgrp", "chmod", "echo", "mkdir", "tee", "touch", "usermod"]

---

**Environment: OS —— Case 17**

Instruction: Create a backup directory "/backup", copy "/source/test.txt" to both "/backup" and "/destination", append "Backup completed" to "/logs/backup.log", set permissions of "/backup/test.txt" to 644, and ensure "/destination/test.txt" is owned by root.

```
Ground Truth Action: mkdir -p /source /destination
/logs && echo "Sample data" > /source/test.txt && touch
/logs/backup.log
```

Skill List: ["chmod", "chown", "cp", "echo", "find", "grep", "ls", "mkdir", "tee"]

---

**Environment: OS —— Case 18**

Instruction: 1. Add "/bin/zsh" to /etc/shells. 2. Change "testuser" login shell to /bin/zsh. 3. Create group "devteam". 4. Add "testuser" to "devteam". 5. Create /shared directory with permissions 770 owned by "devteam" group.

```
Ground Truth Action: addgroup devteam && usermod -aG devteam
testuser && mkdir /shared && chgrp devteam /shared && chmod
770 /shared && echo '/bin/zsh' | tee -a /etc/shells && grep
-q '/bin/zsh' /etc/shells && chsh -s /bin/zsh testuser &&
chage -E 2025-12-31 testuser && touch /home/testuser/.zshrc
&& chown testuser:testuser /home/testuser/.zshrc && echo
'export PATH=/usr/local/bin:\$PATH' >> /home/testuser/.zshrc
```

Skill List: ["addgroup", "chage", "chgrp", "chmod", "chown", "chsh", "echo", "grep", "mkdir", "tee", "touch", "usermod"]

## C.3 KNOWLEDGE GRAPH SAMPLES

---
**Environment: KG —— Case 19**

Instruction: which institution has national wine centre of australia?, Entities: ["National Wine Centre of Australia"]

```
Ground Truth Action:

["get_relations(m.03hd1z)", "get_neighbors(m.03hd1z,
 education.educational_institution.parent_institution)"]
```

Skill List: ["get_neighbors"]

---
**Environment: KG —— Case 20**

Instruction: how many characters appear on the cover of tintin in the land of the soviets?

```
Ground Truth Action:

["get_relations(m.02ll5h)",
 "get_neighbors(m.02ll5h,
 comic_books.comic_book_issue.characters_on_cover)",
 "count(#0)"]
```

Skill List: ["count", "get_neighbors"]

---
**Environment: KG —— Case 21**

Instruction: Question: which short story of the sacred band of stepsons universe universe is know to have the earliest copyright date?, Entities: ["The Sacred Band of Stepsons universe"]

```
Ground Truth Action:

["get_relations(m.0ch8hcq)",
 "get_neighbors(m.0ch8hcq,
 fictional_universe.fictional_universe.works_set_here)",
 "get_attributes(#0)",
 "argmin(#0,book.written_work.copyright_date)"]
```

Skill List: ["argmin", "get_neighbors"]

---
**Environment: KG —— Case 22**

Instruction: which fictional character produced by marv wolfman did trevor von eeden create?, Entities: ["Trevor Von Eeden", "Marv Wolfman"]

```
Ground Truth Action:

["get_relations(m.0279q8n)",
 "get_neighbors(m.0279q8n,
 fictional_universe.fictional_character_creator.
 fictional_characters_created)",
 "get_relations(m.02gn9g)",
 "get_neighbors(m.02gn9g,
 fictional_universe.fictional_character_creator.
 fictional_characters_created)",
 "intersection(#0,#1)"]
```

Skill List: ["get_neighbors", "intersection"]

---

**Environment: KG —— Case 23**

Instruction: what was the most recently formed cyclone in the same category as hurricane dolly?, Entities: ["Hurricane Dolly"]

```
Ground Truth Action:

["get_relations(m.04dn799)",
 "get_neighbors(m.04dn799,
 meteorology.tropical_cyclone.category)",
 "get_relations(#0)",
 "get_neighbors(#0,
 meteorology.tropical_cyclone_category.tropical_cyclones)",
 "get_attributes(#1)",
 "argmax(#1,meteorology.tropical_cyclone.formed)"]
```

Skill List: ["argmax", "get_neighbors"]

---

**Environment: KG —— Case 24**

Instruction: how much content about talk radio is produced by the person that produces weekend edition sunday?, Entities: ["Talk radio", "Weekend Edition Sunday"]

```
Ground Truth Action:

["get_relations(m.07dn1)",
 "get_neighbors(m.07dn1,broadcast.genre.content)",
 "get_relations(m.0t4t10s)",
 "get_neighbors(m.0t4t10s,broadcast.content.producer)",
 "get_relations(#1)",
 "get_neighbors(#1,broadcast.producer.produces)",
 "intersection(#0,#2)",
 "count(#3)"
]
```

Skill List: ["count", "get_neighbors", "intersection"]

---

**Environment: KG —— Case 25**

Instruction: what semi-firm textured cheese is made from the products of lamb and goat?, Entities: ["lamb", "Goat", "semi-firm"]

```
Ground Truth Action:

["get_relations(m.07bgp)",
 "get_neighbors(m.07bgp,food.cheese_milk_source.cheeses)",
 "get_relations(m.03fwl)",
 "get_neighbors(m.03fwl,food.cheese_milk_source.cheeses)",
 "get_relations(m.02h82t0)",
 "get_neighbors(m.02h82t0,food.cheese_texture.cheeses)",
 "intersection(#1,#2)",
 "intersection(#0,#3)"]
```

Skill List: ["get_neighbors", "intersection"]

---

**Environment: KG —— Case 26**

Instruction: is a model of opel super 6 related to a eagle talon?, Entities: ["Opel Super 6", "Eagle Talon"]

```
Ground Truth Action:

["get_relations(m.0gdk70)",
 "get_neighbors(m.0gdk70,automotive.model.automotive_class)",
 "get_relations(#0)",
 "get_neighbors(#0,automotive.automotive_class.examples)",
 "get_relations(m.02p04r)",
 "get_neighbors(m.02p04r,automotive.model.related_models)",
 "get_relations(#2)",
 "get_neighbors(#2,
 automotive.similar_automobile_models.related_model)",
 "intersection(#1,#3)"]
```

Skill List: ["get_neighbors", "intersection"]

---

## D  ERROR CASE ANALYSIS

To better understand the failure modes of LLM agents in LifelongAgentBench , we analyze all task outcomes using two orthogonal attributes: `evaluation_outcome` (correct vs. incorrect) and `sample_status` (detailed agent or system status). This provides a fine-grained categorization of agent behavior and helps identify major bottlenecks.

**Successful Cases (`evaluation_outcome == \correct"`)**  We observe two typical success patterns:

- `sample_status == \completed"`: the agent explicitly submits a valid final answer following the required format (e.g., correctly querying customer feedback with SQL and submitting results).
- `sample_status == \task_limit_reached"`: the agent stops without explicitly committing, but the final database state satisfies the evaluation criteria (e.g., correctly updating records within the step limit).

**Error Cases (`evaluation_outcome == \incorrect"`)**  We identify four common failure modes:

- `sample_status == \completed"`: the agent explicitly commits an incorrect answer despite following the required format (e.g., submitting an incomplete result in a multi-step SQL query).
- `sample_status == \task_limit_reached"`: the agent fails to converge within the interaction limit, often due to redundant or looping operations (e.g., repeatedly adjusting query logic without submitting).
- `sample_status == \agent_validation_failed"`: the agent violates output constraints, such as missing required keywords or formatting (e.g., omitting `Action: Answer` tag).
- `sample_status == \agent_context_limit"`: the agent exceeds the LLM context window, especially in environments requiring long reasoning chains or large intermediate outputs.

**System-Level Failures**  We also observe rare non-agent errors:

- `task_environment_error`: task-side infrastructure failures (e.g., Docker crashes).

- `task_unknown_error`: unknown exceptions in task execution.

- `agent_out_of_memory`: GPU memory exhaustion on large models.

- `agent_unknown_error`: unexpected internal failures within the agent framework.

This taxonomy highlights that even when equipped with powerful LLMs, agents remain vulnerable to subtle evaluation failures, unstable reasoning loops, and external infrastructure issues. Our benchmark thus provides valuable diagnostic insights for developing more robust lifelong learning agents.

## D.1 TASK COMPLETED

### D.1.1 ERROR CASE 1

**Task Description:** The agent was instructed to query the database to retrieve insurance policy details, specifically the policy IDs, holder names, and policy types from the `insurance_policies` table. The results needed to be ordered by `effective_date` from the most recent to the oldest, skipping the first five entries and returning the subsequent ten entries.

**Agent Behavior:** Initially, the agent executed the following SQL query:

```
SELECT policy_id, holder_name, policy_type
FROM insurance_policies
ORDER BY effective_date DESC
LIMIT 4 OFFSET 5;
```

This query was incorrect due to mistakenly using `LIMIT 4` instead of the required `LIMIT 10`. Consequently, the query returned only four entries instead of the intended ten.

Recognizing the mistake after receiving the results, the agent correctly identified the error in the limit clause. It then adjusted and executed the corrected query:

```
SELECT policy_id, holder_name, policy_type
FROM insurance_policies
ORDER BY effective_date DESC
LIMIT 10 OFFSET 5;
```

The updated query returned the correct number of rows (ten entries), satisfying the task requirements.

**Reason for Failure:** Despite eventually correcting the SQL query, the task outcome was marked as incorrect (status `completed` but evaluation incorrect). The failure arose from the initial misunderstanding and execution of an inaccurate query, causing unnecessary computational resources (an extra interaction) and illustrating unstable reasoning. Such errors highlight difficulties in accurately translating natural language constraints (e.g., "starting from the 6th entry and limited to 10 entries") into correct SQL clauses (`OFFSET` and `LIMIT`).

**Implications and Insights:**

- **Unstable Reasoning:** The agent correctly explained the logic but initially failed to accurately apply it, demonstrating inconsistency between stated understanding and execution.

- **Inefficiency:** The error required additional agent-system interactions, revealing suboptimal reasoning efficiency.

- **Potential Solutions:** Future improvements could involve explicit verification of each SQL query component against the original natural language instruction or introducing intermediate verification steps before query execution.

This case exemplifies a common issue where an agent's conceptual interpretation of instructions is correct but its initial execution is flawed, underscoring the necessity for enhanced accuracy in natural language-to-code translation.

Error Case 1 —— Environment: DB —— Error Type: completed

User: I will ask you a question, then you should help me operate a MySQL database with SQL to answer the question.\nYou have to explain the problem and your solution to me and write down your thoughts.\nAfter thinking and explaining thoroughly, every round you can choose one of the two actions: Operation or Answer.\n\nTo do operation, the format should be like this:\nAction: Operation\n"sql\nSELECT  FROM table WHERE condition;\n"\nYou MUST put SQL in markdown format without any other comments. Your SQL should be in one line.\nI will use Äction: Operationïterally to match your SQL.\nEvery time you can only execute one SQL statement. I will only execute the statement in the first SQL code block. Every time you write a SQL, I will execute it for you and give you the output.\nIf the SQL is not executed successfully, the response will be the error message.\nOtherwise, the response will be the raw MySQL response.\nFor SELECT queries, the response will be the result of the query, such as [(1, "John Doe", "HR"), (2, "Jane Smith", "IT"), ...], where each tuple represents a row and the elements are the values of the columns in the row.\nFor SQL such as INSERT, UPDATE, and DELETE, the response will be an empty list [] indicating that the SQL was executed successfully.\n\nIf you have obtain the answer by interacting with the database, and you MUST commit your final answer using the format like this:\nAction: Answer\nFinal Answer: [(1, "John Doe", "HR"), (2, "Jane Smith", "IT"), ...]\nDO NOT write this pattern unless you are sure about your answer. I expect an accurate and correct answer.\nYour answer should be accurate. Your answer must be exactly the same as the correct answer.\nIf the question is about modifying the database, then after done operation, your answer field can be anything.\nIf the question is about querying the database, then after done operation, your answer field should be the result of the query.\nWe note that the column names will not be displayed in the result, and you need to ensure both the orders of the columns and rows are correct.\nIf your response cannot match any pattern I mentioned earlier, you will be judged as FAIL immediately.\nOnce you commit your answer or the number of rounds reaches 3, the task will be finished and the system will judge whether you pass the task or not.\n\nNow, I will give you the question that you need to solve.

Agent: OK.

User: What are the policy IDs, holder names, and policy types of the insurance policies? Return the results ordered by effective date from most recent to oldest, starting from the 6th entry and limited to 10 entries.\nThe name of this table is insurance_policies, and the headers of this table are policy_id, holder_name, policy_type, effective_date, premium_amount, coverage_limit.

Agent: To solve this problem, we need to first understand the requirements. We need to return the policy IDs, holder names, and policy types of the insurance policies. We also need to order the results by effective date from most recent to oldest, starting from the 6th entry and limited to 10 entries.\n\nHere's my thought process:\n\n1. We need to select the policy ID, holder name, and policy type from the insurance_policies table.\n2. We need to order the results by effective date in descending order (most recent to oldest).\n3. We need to start the result set from the 6th entry.\n4. We need to limit the result set to 10 entries.\n\nBased on this, I will choose the Öperationäction to execute the SQL query.\n\nAction: Operation\n"sql\nSELECT policy_id, holder_name, policy_type \nFROM insurance_policies \nORDER BY effective_date DESC \nLIMIT 4 OFFSET 5;\n"\nPlease execute this SQL query.

User: [(6, "Alice Brown", "Health"), (7, "Bob Green", "Auto"), (8, "Charlie White", "Life"), (9, "Diana Black", "Home")]

Agent: The query I executed returned the first 4 rows from the 6th row onwards, ordered by effective date in descending order. However, this is not what we want. We want to return the first 10 rows from the 6th row onwards, ordered by effective date in descending order.\n\nTo achieve this, we need to modify the query to return the first 10 rows from the 6th row onwards, ordered by effective date in descending order.\n\nHere's my thought

process:\n\n1. We need to select the policy ID, holder name, and policy type from the insurance_policies table.\n2. We need to order the results by effective date in descending order (most recent to oldest).\n3. We need to start the result set from the 6th entry.\n4. We need to limit the result set to 10 entries.\n\nBased on this, I will choose the Öperationäction to execute the SQL query.\n\nAction: Operation\n"sql\nSELECT policy_id, holder_name, policy_type \nFROM insurance_policies \nORDER BY effective_date DESC \nLIMIT 10 OFFSET 5;\n"

User: [(6, "Alice Brown", "Health"), (7, "Bob Green", "Auto"), (8, "Charlie White", "Life"), (9, "Diana Black", "Home"), (10, "Eve Gray", "Travel"), (11, "Alice Brown", "Health"), (12, "Bob Green", "Auto"), (13, "Charlie White", "Life"), (14, "Diana Black", "Home"), (15, "Eve Gray", "Travel")]

Agent: The query I executed returned the first 10 rows from the 6th row onwards, ordered by effective date in descending order. This is what we want.\n\nNow, we can commit the final answer.\n\nAction: Answer\nFinal Answer: [(6, "Alice Brown", "Health"), (7, "Bob Green", "Auto"), (8, "Charlie White", "Life"), (9, "Diana Black", "Home"), (10, "Eve Gray", "Travel"), (11, "Alice Brown", "Health"), (12, "Bob Green", "Auto"), (13, "Charlie White", "Life"), (14, "Diana Black", "Home"), (15, "Eve Gray", "Travel")]

### D.1.2 ERROR CASE 2

**Task Description:** The agent was tasked with querying the database to identify regions from the `customer_usage` table that have a total data usage exceeding 1000 and an average call duration under 300. The results needed to include the region, total data usage, and average call duration, grouped by region, and limited to 5 entries starting from the third entry.

**Agent Behavior:** Initially, the agent executed the following SQL query:

```
SELECT region, SUM(data_usage) AS total_data_usage,
AVG(call_duration) AS average_call_duration
FROM customer_usage
GROUP BY region
HAVING SUM(data_usage) > 1000 AND AVG(call_duration) < 300
ORDER BY total_data_usage DESC
LIMIT 2, 5;
```

However, the agent misunderstood the usage of `LIMIT` and `OFFSET` syntax in SQL, using `LIMIT 2, 5` instead of the correct `LIMIT 5 OFFSET 2` syntax. After executing the initial query, the agent attempted to correct its misunderstanding by issuing a second query:

```
SELECT region, SUM(data_usage) AS total_data_usage,
AVG(call_duration) AS average_call_duration
FROM customer_usage
GROUP BY region
HAVING SUM(data_usage) > 1000 AND AVG(call_duration) < 300
ORDER BY total_data_usage DESC
LIMIT 3, 5;
```

Yet, this second query also incorrectly interpreted the desired offset and limit, resulting in incorrect indexing.

**Reason for Failure:** The agent's failure stemmed from confusion regarding the SQL syntax for specifying offsets and limits. Instead of correctly applying `OFFSET 2 LIMIT 5` to start from the third entry, the agent incorrectly applied `LIMIT 2,5` and subsequently `LIMIT 3,5`, leading to incorrect or shifted result sets.

**Implications and Insights:**

- **Syntax Misinterpretation:** The agent demonstrated difficulty accurately applying SQL syntax for offset and limit clauses, highlighting issues in translating natural language queries to precise SQL statements.

- **Persistent Mistake:** Despite attempting to correct its mistake, the agent continued to misuse the syntax, illustrating repeated misunderstanding rather than one-off oversight.

- **Potential Solutions:** Improved SQL syntax awareness or explicit syntactic verification mechanisms during intermediate reasoning steps could significantly enhance accuracy.

This case highlights the importance of reinforcing SQL syntax knowledge and accuracy during the reasoning and execution phases for LLM agents.

---

Error Case 2 —— Environment: DB —— Error Type: completed

User: I will ask you a question, then you should help me operate a MySQL database with SQL to answer the question.\nYou have to explain the problem and your solution to me and write down your thoughts.\nAfter thinking and explaining thoroughly, every round you can choose one of the two actions: Operation or Answer.\n\nTo do operation, the format should be like this:\nAction: Operation\n"sql\nSELECT FROM table WHERE condition;\n"\nYou MUST put SQL in markdown format without any other comments. Your SQL should be in one line.\nI will use "Action: Operation" literally to match your SQL.\nEvery time you can only execute one SQL statement. I will only execute the statement in the first SQL code block. Every time you write a SQL, I will execute it for you and give you the output.\nIf the SQL is not executed successfully, the response will be the error message.\nOtherwise, the response will be the raw MySQL response.\nFor SELECT queries, the response will be the result of the query, such as [(1, "John Doe", "HR"), (2, "Jane Smith", "IT"), ...], where each tuple represents a row and the elements are the values of the columns in the row.\nFor SQL such as INSERT, UPDATE, and DELETE, the response will be an empty list [] indicating that the SQL was executed successfully.\n\nIf you have obtain the answer by interacting with the database, and you MUST commit your final answer using the format like this:\nAction: Answer\nFinal Answer: [(1, "John Doe", "HR"), (2, "Jane Smith", "IT"), ...]\nDO NOT write this pattern unless you are sure about your answer. I expect an accurate and correct answer.\nYour answer should be accurate. Your answer must be exactly the same as the correct answer.\nIf the question is about modifying the database, then after done operation, your answer field can be anything.\nIf the question is about querying the database, then after done operation, your answer field should be the result of the query.\nWe note that the column names will not be displayed in the result, and you need to ensure both the orders of the columns and rows are correct.\nIf your response cannot match any pattern I mentioned earlier, you will be judged as FAIL immediately.\nOnce you commit your answer or the number of rounds reaches 3, the task will be finished and the system will judge whether you pass the task or not.\n\nNow, I will give you the question that you need to solve.

Agent: OK.

User: Which regions have a total data usage exceeding 1000 and an average call duration under 300? Return the region, total data usage, and average call duration, grouped by region. Limit the results to 5 entries starting from the third entry.\nThe name of this table is customer_usage, and the headers of this table are customer_id, region, data_usage, call_duration, month.

Agent: To solve this problem, we need to first calculate the total data usage and average call duration for each region. Then, we can filter the results to only include regions with a total data usage exceeding 1000 and an average call duration under 300. Finally, we can group the results by region and limit them to 5 entries starting from the third entry.\n\nAction: Operation\n"sql\nSELECT region, SUM(data_usage) AS total_data_usage, AVG(call_duration) AS average_call_duration \nFROM customer_usage \nGROUP BY region \nHAVING SUM(data_usage) ¿ 1000 AND AVG(call_duration) ¡ 300 \nORDER BY total_data_usage DESC \nLIMIT 2, 5;\n"

User: [("South", Decimal("1200"), Decimal("250.0000")), ("Central", Decimal("1200"), Decimal("280.0000")), ("EastCoast", Decimal("1100"), Decimal("290.0000")), ("WestCoast", Decimal("1100"), Decimal("200.0000")), ("Southwest", Decimal("1050"), Decimal("290.0000"))]

Agent: The results show the top 5 regions with a total data usage exceeding 1000 and an average call duration under 300. However, we need to exclude the first two entries since we want to start from the third entry.\n\nAction: Operation\n"sql\nSELECT region, SUM(data_usage) AS total_data_usage, AVG(call_duration) AS average_call_duration \nFROM customer_usage \nGROUP BY region \nHAVING SUM(data_usage) ¿ 1000 AND AVG(call_duration) ¡ 300 \nORDER BY total_data_usage DESC \nLIMIT 3, 5;\n"

User: [("Central", Decimal("1200"), Decimal("280.0000")), ("EastCoast", Decimal("1100"), Decimal("290.0000")), ("WestCoast", Decimal("1100"), Decimal("200.0000")), ("Southwest", Decimal("1050"), Decimal("290.0000"))]

Agent: The results now show the top 5 regions with a total data usage exceeding 1000 and an average call duration under 300, starting from the third entry.\n\nAction: Answer\nFinal Answer: [("Central", Decimal("1200"), Decimal("280.0000")), ("EastCoast", Decimal("1100"), Decimal("290.0000")), ("WestCoast", Decimal("1100"), Decimal("200.0000")), ("Southwest", Decimal("1050"), Decimal("290.0000"))]

## D.2 TASK LIMITED REACHED

### D.2.1 ERROR CASE 3

**Task Description:** The agent was tasked with updating the hourly rates for staff members in Security roles from the `staff_shifts` table. Specifically, it was required to increase the hourly rate by 15% for Security roles where the hours worked exceeded 30, and the current hourly rate was below the average hourly rate of all Security roles.

**Agent Behavior:** Initially, the agent correctly calculated the average hourly rate for Security roles using the following SQL query:

```
SELECT AVG(hourly_rate) AS avg_rate
FROM staff_shifts
WHERE role = "Security";
```

Upon identifying the average hourly rate (20.5), the agent correctly queried the eligible entries to be updated:

```
SELECT * FROM staff_shifts
WHERE role = "Security" AND hours_worked > 30 AND hourly_rate < 20.5;
```

Having successfully retrieved the targeted entries, the agent executed the update query:

```
UPDATE staff_shifts
SET hourly_rate = hourly_rate * 1.15
WHERE role = "Security" AND hours_worked > 30
AND hourly_rate < 20.5;
```

Finally, the agent attempted to verify the successful execution of the update by recalculating the average hourly rate. However, at this point, the agent reached the maximum allowed interaction steps, failing to explicitly commit a final answer.

**Reason for Failure:** The agent failed due to exceeding the maximum interaction limit without committing a final answer explicitly. Although the operations performed were logically sound and correct, the agent did not conclude the task by confirming the success explicitly within the allowed steps.

**Implications and Insights:**

- **Lack of Task Commitment:** The agent demonstrated proper initial reasoning and executed correct SQL statements. However, it failed to finalize the task promptly, reflecting inadequate task closure behavior.

- **Interaction Efficiency:** This case highlights the necessity for the agent to strategically plan and optimize interactions, especially when task constraints (such as interaction limits) are strict.

- **Potential Solutions:** Introducing intermediate commitment strategies or improved planning to manage interaction limits effectively could enhance overall task performance and success rates.

This case emphasizes the importance of explicit task completion strategies and efficient interaction management in multi-step tasks for LLM agents.

---

Error Case 3 —— Environment: DB —— Error Type: task limit reached

User: I will ask you a question, then you should help me operate a MySQL database with SQL to answer the question.\nYou have to explain the problem and your solution to me and write down your thoughts.\nAfter thinking and explaining thoroughly, every round you can choose one of the two actions: Operation or Answer.\n\nTo do operation, the format should be like this:\nAction: Operation\n"sql\nSELECT  FROM table WHERE condition;\n"\nYou MUST put SQL in markdown format without any other comments. Your SQL should be in one line.\nI will use "Action: Operation" literally to match your SQL.\nEvery time you can only execute one SQL statement. I will only execute the statement in the first SQL code block. Every time you write a SQL, I will execute it for you and give you the output.\nIf the SQL is not executed successfully, the response will be the error message.\nOtherwise, the response will be the raw MySQL response.\nFor SELECT queries, the response will be the result of the query, such as [(1, "John Doe", "HR"), (2, "Jane Smith", "IT"), ...], where each tuple represents a row and the elements are the values of the columns in the row.\nFor SQL such as INSERT, UPDATE, and DELETE, the response will be an empty list [] indicating that the SQL was executed successfully.\n\nIf you have obtain the answer by interacting with the database, and you MUST commit your final answer using the format like this:\nAction: Answer\nFinal Answer: [(1, "John Doe", "HR"), (2, "Jane Smith", "IT"), ...]\nDO NOT write this pattern unless you are sure about your answer. I expect an accurate and correct answer.\nYour answer should be accurate. Your answer must be exactly the same as the correct answer.\nIf the question is about modifying the database, then after done operation, your answer field can be anything.\nIf the question is about querying the database, then after done operation, your answer field should be the result of the query.\nWe note that the column names will not be displayed in the result, and you need to ensure both the orders of the columns and rows are correct.\nIf your response cannot match any pattern I mentioned earlier, you will be judged as FAIL immediately.\nOnce you commit your answer or the number of rounds reaches 3, the task will be finished and the system will judge whether you pass the task or not.\n\nNow, I will give you the question that you need to solve.

Agent: OK.

User: Update the hourly rate by increasing it by 15% for Security roles where hours worked exceed 30 and their current hourly rate is below the average hourly rate of all Security roles.\nThe name of this table is staff_shifts, and the headers of this table are shift_id, staff_name, role, shift_date, hours_worked, hourly_rate, total_pay.

Agent: To solve this problem, we need to first calculate the average hourly rate of all Security roles. Then, we need to find the Security roles where hours worked exceed 30 and their current hourly rate is below the average hourly rate of all Security roles. Finally, we can update the hourly rate of these roles by increasing it by 15%.\n\nAction: Operation\n"sql\nSELECT AVG(hourly_rate) AS avg_rate FROM staff_shifts WHERE role = "Security";\n"

---

User: [(Decimal("20.5000"),)]

Agent: The average hourly rate of all Security roles is 20.5. Now, we need to find the Security roles where hours worked exceed 30 and their current hourly rate is below 20.5.\n\nAction: Operation\n"sql\nSELECT  FROM staff_shifts WHERE role = "Security" AND hours_worked ¿ 30 AND hourly_rate ¡ 20.5;\n"

User: [(1, "Mike Johnson", "Security", "2023-01-01", 35, 18, 630), (2, "Anna Lee", "Security", "2023-01-02", 32, 20, 640), (5, "Evan Smith", "Security", "2023-01-05", 31, 19, 589), (7, "George Adams", "Security", "2023-01-07", 33, 17, 561), (10, "Jenna Hall", "Security", "2023-01-10", 34, 16, 544)]

Agent: We have found the Security roles that meet the conditions. Now, we can update the hourly rate of these roles by increasing it by 15%.\n\nAction: Operation\n"sql\nUPDATE staff_shifts SET hourly_rate = hourly_rate  1.15 WHERE role = "Security" AND hours_worked ¿ 30 AND hourly_rate ¡ 20.5;\n"

User: []

Agent: The update operation was successful. Now, we need to recalculate the average hourly rate of all Security roles to confirm that the update operation was correct.\n\nAction: Operation\n"sql\nSELECT AVG(hourly_rate) AS avg_rate FROM staff_shifts WHERE role = "Security";\n"

## D.3 ERROR CASE 4

**Task Description:** The agent was tasked with querying the `tax_records` table to identify tax statuses and their total amounts owed for the tax year 2023, specifically focusing on statuses labeled `\unpaid"` or `\pending"`. The results needed to be grouped by status, ordered by status in descending order, and within each status, by total owed in ascending order.

**Agent Behavior:** Initially, the agent executed the following correct SQL query:

```
SELECT status, SUM(amount_owed) as total_owed
FROM tax_records
WHERE tax_year = 2023 AND status IN ("unpaid", "pending")
GROUP BY status
ORDER BY status DESC, total_owed ASC;
```

The query returned the desired result set correctly:

```
[("unpaid", Decimal("6800")), ("pending", Decimal("6100"))]
```

However, the agent incorrectly assumed that additional sorting logic was needed to handle cases with identical total owed amounts, even though no such cases were present. This misunderstanding led to multiple redundant attempts using SQL functions like `FIELD()` and conditional expressions (`IF()`) that did not alter the result. The agent unnecessarily repeated query attempts without reaching a conclusion.

**Reason for Failure:** The agent failed due to repeatedly performing redundant and unnecessary operations that did not alter the query outcome, exhausting the maximum allowed interaction steps without explicitly committing the final correct answer.

**Implications and Insights:**

- **Redundant Reasoning:** The agent's repeated attempts to address an imagined problem illustrate ineffective recognition of successfully completed tasks, leading to redundant queries.

- **Misinterpretation of Results:** Despite correct initial execution, the agent incorrectly inferred a need for additional complex sorting logic, indicating a misunderstanding of task completion criteria.

- **Potential Solutions:** Encouraging explicit checks or verification steps to recognize already successful outcomes could help avoid redundant actions. Improved monitoring of agent understanding could also mitigate unnecessary complexity.

This case emphasizes the necessity of accurate result interpretation and effective termination strategies to prevent redundant interactions in multi-step tasks for LLM agents.

---

Error Case 4 —— Environment: DB —— Error Type: task_limit_reached

User: I will ask you a question, then you should help me operate a MySQL database with SQL to answer the question.\nYou have to explain the problem and your solution to me and write down your thoughts.\nAfter thinking and explaining thoroughly, every round you can choose one of the two actions: Operation or Answer.\n\nTo do operation, the format should be like this:\nAction: Operation\n"sql\nSELECT  FROM table WHERE condition;\n"\nYou MUST put SQL in markdown format without any other comments. Your SQL should be in one line.\nI will use "Action: Operation" literally to match your SQL.\nEvery time you can only execute one SQL statement. I will only execute the statement in the first SQL code block. Every time you write a SQL, I will execute it for you and give you the output.\nIf the SQL is not executed successfully, the response will be the error message.\nOtherwise, the response will be the raw MySQL response.\nFor SELECT queries, the response will be the result of the query, such as [(1, "John Doe", "HR"), (2, "Jane Smith", "IT"), ...], where each tuple represents a row and the elements are the values of the columns in the row.\nFor SQL such as INSERT, UPDATE, and DELETE, the response will be an empty list [] indicating that the SQL was executed successfully.\n\nIf you have obtain the answer by interacting with the database, and you MUST commit your final answer using the format like this:\nAction: Answer\nFinal Answer: [(1, "John Doe", "HR"), (2, "Jane Smith", "IT"), ...]\nDO NOT write this pattern unless you are sure about your answer. I expect an accurate and correct answer.\nYour answer should be accurate. Your answer must be exactly the same as the correct answer.\nIf the question is about modifying the database, then after done operation, your answer field can be anything.\nIf the question is about querying the database, then after done operation, your answer field should be the result of the query.\nWe note that the column names will not be displayed in the result, and you need to ensure both the orders of the columns and rows are correct.\nIf your response cannot match any pattern I mentioned earlier, you will be judged as FAIL immediately.\nOnce you commit your answer or the number of rounds reaches 3, the task will be finished and the system will judge whether you pass the task or not.\n\nNow, I will give you the question that you need to solve.

Agent: OK.

User: What are the statuses and their total amounts owed for the tax year 2023 where the status is either "unpaid" or "pending"? Return each status and the corresponding total amount owed, grouped by status, ordered by status in descending order and total owed in ascending order.\nThe name of this table is tax_records, and the headers of this table are taxpayer_id, name, tax_year, amount_owed, status, filing_date.

Agent: To solve this problem, we need to first identify the tax records for the tax year 2023 where the status is either "unpaid" or "pending". Then, we can group these records by status and calculate the total amount owed for each status. Finally, we can order the results by status in descending order and total owed in ascending order.\n\nHere's my thought process:\n\n1. We need to select the tax records for the tax year 2023 where the status is either "unpaid" or "pending".\n2. We need to group these records by status and calculate the total amount owed for each status.\n3. We need to order the results by status in descending order and total owed in ascending order.\n\nBased on these steps, I will choose the "Operation" action to execute the SQL query.\n\nAction: Operation\n"sql\nSELECT status, SUM(amount_owed) as total_owed FROM tax_records WHERE tax_year = 2023 AND

---

status IN (”unpaid“, ”pending“) GROUP BY status ORDER BY status DESC, total_owed ASC;\n”

User: [(“unpaid”, Decimal(“6800”)), (“pending”, Decimal(“6100”))]

Agent: The query returned the total amount owed for each status. However, we also need to consider the case where the total amount owed is the same for two or more statuses. In this case, we want to order the statuses in descending order, and then order the statuses with the same total amount owed in ascending order of their status names.\n\nTo achieve this, we can modify the query to use the FIELD function, which returns the index of the first occurrence of a value in a list. We can use this function to order the statuses in descending order, and then order the statuses with the same total amount owed in ascending order of their status names.\n\nHere's my thought process:\n\n1. We need to select the tax records for the tax year 2023 where the status is either “unpaid” or “pending”.\n2. We need to group these records by status and calculate the total amount owed for each status.\n3. We need to order the results by status in descending order, and then order the statuses with the same total amount owed in ascending order of their status names.\n\nBased on these steps, I will choose the “Operation” action to execute the SQL query.\n\nAction: Operation\n“sql\nSELECT status, SUM(amount_owed) as total_owed FROM tax_records WHERE tax_year = 2023 AND status IN (”unpaid“, ”pending“) GROUP BY status ORDER BY FIELD(status, ”unpaid“, ”pending“), total_owed ASC;\n”

User: [(“unpaid”, Decimal(“6800”)), (“pending”, Decimal(“6100”))]

Agent: The query still didn't order the statuses with the same total amount owed in ascending order of their status names. To achieve this, we can use a combination of the FIELD function and the IF function to create a custom ordering.\n\nHere's my thought process:\n\n1. We need to select the tax records for the tax year 2023 where the status is either “unpaid” or “pending”.\n2. We need to group these records by status and calculate the total amount owed for each status.\n3. We need to order the results by status in descending order, and then order the statuses with the same total amount owed in ascending order of their status names.\n\nBased on these steps, I will choose the “Operation” action to execute the SQL query.\n\nAction: Operation\n“sql\nSELECT status, SUM(amount_owed) as total_owed FROM tax_records WHERE tax_year = 2023 AND status IN (”unpaid“, ”pending“) GROUP BY status ORDER BY FIELD(status, ”unpaid“, ”pending“), IF(status = ”unpaid“, 0, 1), total_owed ASC;\n”

User: [(“unpaid”, Decimal(“6800”)), (“pending”, Decimal(“6100”))]

Agent: The query still didn't order the statuses with the same total amount owed in ascending order of their status names. To achieve this, we can use a combination of the FIELD function and the IF function to create a custom ordering.\n\nHowever, I realized that the previous queries were close, but the FIELD function was not necessary. We can simply use the status names in the ORDER BY clause to achieve the desired ordering.\n\nHere's my thought process:\n\n1. We need to select the tax records for the tax year 2023 where the status is either “unpaid” or “pending”.\n2. We need to group these records by status and calculate the total amount owed for each status.\n3. We need to order the results by status in descending order, and then order the statuses with the same total amount owed in ascending order of their status names.\n\nBased on these steps, I will choose the “Operation” action to execute the SQL query.\n\nAction: Operation\n“sql\nSELECT status, SUM(amount_owed) as total_owed FROM tax_records WHERE tax_year = 2023 AND status IN (”unpaid“, ”pending“) GROUP BY status ORDER BY status DESC, total_owed ASC;\n”

### D.4 AGENT VALIDATION FAILED

#### D.4.1 ERROR CASE 5

**Task Description:** The agent was tasked with deleting up to 100 records from the `package_tracking` table, specifically targeting packages marked as \lost" with a last update before January 1, 2023, or packages weighing less than 5 kilograms.

**Agent Behavior:** Initially, the agent executed an incorrect SQL query attempting to combine `LIMIT` with a subquery:

```
DELETE FROM package_tracking
WHERE (status = "lost" AND last_updated < "2023-01-01")
OR package_id IN (
  SELECT package_id FROM package_tracking WHERE weight_kg < 5 LIMIT 100
);
```

This resulted in a MySQL syntax error, as `LIMIT` is not supported within subqueries in MySQL.

Recognizing this error, the agent attempted a valid alternative by using a temporary table to bypass the limitation:

```
CREATE TEMPORARY TABLE temp AS
SELECT package_id FROM package_tracking WHERE weight_kg < 5 LIMIT 100;

DELETE FROM package_tracking
WHERE (status = "lost" AND last_updated < "2023-01-01")
OR package_id IN (SELECT package_id FROM temp);
```

Although logically correct, the agent mistakenly executed two SQL statements simultaneously, violating the instructions that explicitly require executing only one SQL statement per interaction.

**Reason for Failure:** The task failed due to an `agent_validation_failed` error, specifically because the agent violated the given operational constraints by submitting multiple SQL statements in one operation instead of executing them sequentially.

**Implications and Insights:**

- **Instruction Adherence:** The agent clearly understood the logical solution but overlooked explicit instructions requiring single-statement execution per interaction.
- **Validation Awareness:** This highlights the need for the agent to better adhere to clearly defined operational constraints, even when the logical reasoning is correct.
- **Potential Solutions:** Implementing explicit intermediate validation checks or enhancing agent instruction comprehension could significantly reduce such format violations.

This case illustrates the importance of strict adherence to operational instructions and suggests reinforcing agent compliance through validation checks to prevent procedural errors.

---

Error Case 5 —— Environment: DB —— Error Type: agent_validation_failed

User: I will ask you a question, then you should help me operate a MySQL database with SQL to answer the question.\nYou have to explain the problem and your solution to me and write down your thoughts.\nAfter thinking and explaining thoroughly, every round you can choose one of the two actions: Operation or Answer.\n\nTo do operation, the format should be like this:\nAction: Operation\n"sql\nSELECT FROM table WHERE condition;\n"\nYou MUST put SQL in markdown format without any other comments. Your SQL should be in one line.\nI will use "Action: Operation" literally to match your SQL.\nEvery time you can only execute one SQL statement. I will only execute the statement in the first SQL code block. Every time you write a SQL, I will execute it for you and give you the output.\nIf the SQL is not executed successfully, the response will be the error message.\nOtherwise, the response will be the raw MySQL response.\nFor SELECT queries, the response will be the result of the query, such as [(1, "John Doe", "HR"), (2, "Jane Smith",

---

"IT"), ...], where each tuple represents a row and the elements are the values of the columns in the row.\nFor SQL such as INSERT, UPDATE, and DELETE, the response will be an empty list [] indicating that the SQL was executed successfully.\n\nIf you have obtain the answer by interacting with the database, and you MUST commit your final answer using the format like this:\nAction: Answer\nFinal Answer: [(1, "John Doe", "HR"), (2, "Jane Smith", "IT"), ...]\nDO NOT write this pattern unless you are sure about your answer. I expect an accurate and correct answer.\nYour answer should be accurate. Your answer must be exactly the same as the correct answer.\nIf the question is about modifying the database, then after done operation, your answer field can be anything.\nIf the question is about querying the database, then after done operation, your answer field should be the result of the query.\nWe note that the column names will not be displayed in the result, and you need to ensure both the orders of the columns and rows are correct.\nIf your response cannot match any pattern I mentioned earlier, you will be judged as FAIL immediately.\nOnce you commit your answer or the number of rounds reaches 3, the task will be finished and the system will judge whether you pass the task or not.\n\n\nQuestion What are the pet IDs, checkup dates, and weights in kilograms for checkups with a discharge status of "Completed"? Return the results ordered by checkup date in ascending order and then by weight in ascending order.\nThe name of this table is animal_checkups, and the headers of this table are checkup_id, pet_id, checkup_date, weight_kg, veterinarian_id, discharge_status.:\nassistant: To solve this problem, we need to filter the checkups with a discharge status of "Completed" and then order the results by checkup date in ascending order and then by weight in ascending order.\n\nAction: Operation\n"sql\nSELECT pet_id, checkup_date, weight_kg FROM animal_checkups WHERE discharge_status = "Completed" ORDER BY checkup_date ASC, weight_kg ASC;\n"\nuser: [(2, "2020-03-15", 4), (1, "2020-03-15", 5), (1, "2020-07-22", 6), (4, "2021-05-20", 7), (3, "2021-05-20", 8), (6, "2022-01-10", 9), (5, "2022-01-10", 10), (7, "2022-06-18", 12), (10, "2023-04-05", 13), (9, "2023-04-05", 14), (8, "2023-04-05", 15), (5, "2023-10-30", 20)]\nassistant: Action: Answer\nFinal Answer: [(2, "2020-03-15", 4), (1, "2020-03-15", 5), (1, "2020-07-22", 6), (4, "2021-05-20", 7), (3, "2021-05-20", 8), (6, "2022-01-10", 9), (5, "2022-01-10", 10), (7, "2022-06-18", 12), (10, "2023-04-05", 13), (9, "2023-04-05", 14), (8, "2023-04-05", 15), (5, "2023-10-30", 20)]\n\nNow, I will give you the question that you need to solve.

Agent: OK.

User: Delete up to 100 packages that are either marked as "lost" with a last update before January 1, 2023, or have an ID that belongs to packages weighing less than 5 kilograms.\nThe name of this table is package_tracking, and the headers of this table are package_id, status, last_updated, weight_kg, destination_zip, carrier_code, estimated_delivery.

Agent: To solve this problem, we need to filter the packages that are either marked as "lost" with a last update before January 1, 2023, or have an ID that belongs to packages weighing less than 5 kilograms, and then delete up to 100 of these packages.\n\nAction: Operation\n"sql\nDELETE FROM package_tracking WHERE (status = "lost" AND last_updated ¡ "2023-01-01") OR package_id IN (SELECT package_id FROM package_tracking WHERE weight_kg ¡ 5 LIMIT 100);\n"

User: 1235 (42000): This version of MySQL doesn't yet support "LIMIT & IN/ALL/ANY/SOME subquery"

Agent: This error is due to the fact that the LIMIT clause cannot be used in the subquery. We can use a temporary table to solve this problem.\n\nAction: Operation\n"sql\nCREATE TEMPORARY TABLE temp AS SELECT package_id FROM package_tracking WHERE weight_kg ¡ 5 LIMIT 100;\nDELETE FROM package_tracking WHERE (status = "lost" AND last_updated ¡ "2023-01-01") OR package_id IN (SELECT package_id FROM temp);\n"

User: []

Agent: Action: Answer\nThe operation was successful, and up to 100 packages were deleted.

### D.4.2 ERROR CASE 6

**Task Description:** The agent was instructed to insert a new payment record into the `membership_payments` table for member ID 102, specifying the payment date ("2023-10-15"), amount (75), and payment method ("Credit Card").

**Agent Behavior:** The agent attempted the following SQL operation:

```
INSERT INTO membership_payments (member_id, payment_date, amount, payment_method)
VALUES (102, "2023-10-15", 75, "Credit Card");
```

However, the agent explicitly violated the instructions by failing to wrap the SQL command within the required markdown-formatted code block. According to the task instructions, the SQL statement must strictly follow the specified markdown format and execution structure.

**Reason for Failure:** The task failed due to an `agent_validation_failed` error resulting from a violation of the explicit formatting instructions. The agent executed the correct logical SQL command but neglected to encapsulate it correctly within markdown syntax.

**Implications and Insights:**

- **Instruction Compliance:** The agent's oversight highlights a gap in strict adherence to specified formatting and structural guidelines, even when logical accuracy is maintained.

- **Attention to Detail:** This example underscores the necessity for careful attention to format-specific constraints, which are critical for system validation purposes.

- **Potential Solutions:** Enhancing the agent's ability to rigorously follow explicit formatting rules or implementing automated format checks before submission could prevent such failures.

This case emphasizes the critical role of meticulous compliance with explicitly stated operational instructions, highlighting the need for enhanced format verification mechanisms within agent interactions.

---

Error Case 6 —— Environment: DB —— Error Type: agent_validation_failed

User: I will ask you a question, then you should help me operate a MySQL database with SQL to answer the question.\nYou have to explain the problem and your solution to me and write down your thoughts.\nAfter thinking and explaining thoroughly, every round you can choose one of the two actions: Operation or Answer.\n\nTo do operation, the format should be like this:\nAction: Operation\n"sql\nSELECT  FROM table WHERE condition;\n"\nYou MUST put SQL in markdown format without any other comments. Your SQL should be in one line.\nI will use "Action: Operation" literally to match your SQL.\nEvery time you can only execute one SQL statement. I will only execute the statement in the first SQL code block. Every time you write a SQL, I will execute it for you and give you the output.\nIf the SQL is not executed successfully, the response will be the error message.\nOtherwise, the response will be the raw MySQL response.\nFor SELECT queries, the response will be the result of the query, such as [(1, "John Doe", "HR"), (2, "Jane Smith", "IT"), ...], where each tuple represents a row and the elements are the values of the columns in the row.\nFor SQL such as INSERT, UPDATE, and DELETE, the response will be an empty list [] indicating that the SQL was executed successfully.\n\nIf you have obtain the answer by interacting with the database, and you MUST commit your final answer using the format like this:\nAction: Answer\nFinal Answer: [(1, "John Doe", "HR"), (2, "Jane Smith", "IT"), ...]\nDO NOT write this pattern unless you are sure about your answer. I expect an accurate and correct answer.\nYour answer should be accurate. Your answer must be exactly the same as the correct answer.\nIf the question is about modifying the database, then after done operation, your answer field can be anything.\nIf the question is about querying the

> database, then after done operation, your answer field should be the result of the query.\nWe note that the column names will not be displayed in the result, and you need to ensure both the orders of the columns and rows are correct.\nIf your response cannot match any pattern I mentioned earlier, you will be judged as FAIL immediately.\nOnce you commit your answer or the number of rounds reaches 3, the task will be finished and the system will judge whether you pass the task or not.\n\nNow, I will give you the question that you need to solve.
>
> Agent: OK.
>
> User: Insert a new payment record for member ID 102 with a payment date of "2023-10-15", amount 75, and payment method "Credit Card" into the membership payments table.\nThe name of this table is membership_payments, and the headers of this table are payment_id, member_id, payment_date, amount, payment_method.
>
> Agent: Action: Operation\nINSERT INTO membership_payments (member_id, payment_date, amount, payment_method) VALUES (102, "2023-10-15", 75, "Credit Card");

### D.5 AGENT CONTEXT LIMIT

#### D.5.1 ERROR CASE 7

**Task Description:** The agent was given a system administration task on an Ubuntu system: to append several outputs and messages (including static strings, directory contents, hostname, and extracted file data) to the file /var/log/tee_test.log using the tee command, and then set the file's permissions to 644 and ownership to root:adm. The task had a strict interaction cap of five rounds.

**Agent Behavior:** The agent provided a correct breakdown of the required steps and formulated a Bash script attempting to execute all operations in one go:

```
tee -a /var/log/tee_test.log <<< "Line 1"
tee -a /var/log/tee_test.log <<< "Line 2"
find / -print | tee -a /var/log/tee_test.log
hostname | tee -a /var/log/tee_test.log
grep root /etc/passwd | tee -a /var/log/tee_test.log
wc -l /etc/passwd | tee -a /var/log/tee_test.log
tee -a /var/log/tee_test.log <<< "Line 3"
tee -a /var/log/tee_test.log <<< "Done"
chmod 644 /var/log/tee_test.log
chown root:adm /var/log/tee_test.log
```

Although all command steps were logically appropriate, the inclusion of the find / command caused the output to explode in size, producing over 4 million characters of output and ultimately exceeding the LLM context window.

**Reason for Failure:** The task failed due to the agent_context_limit error. The excessive length of the command output (especially from find /) overwhelmed the language model's context window, causing the agent to become unable to continue interaction or process the remainder of the task.

**Implications and Insights:**

- **Unbounded Output Risks:** The use of high-volume commands like find / without filtering or scoping demonstrates a lack of foresight about output size constraints.
- **Context Management Deficiency:** This case highlights the need for LLM agents to proactively assess output volume when planning shell operations in multi-step tasks.
- **Potential Solutions:** Agents should be trained or prompted to use output-limiting constructs (e.g., find / -maxdepth 1, head, du -sh) or to chunk operations across interactions to preserve context capacity.

2430 This case underscores a crucial limitation of LLM agents when interacting with open-ended system
2431 environments: the inability to manage large outputs effectively. It motivates future work on context-
2432 aware execution planning and resource-aware decision making for long-context tasks.
2433
2434
2435
2436
2437

---

Error Case 7 —— Environment: OS —— Error Type: agent_context_limit

2438 User: I will provide you with a task to perform on a Linux (Ubuntu) system. Your objective
2439 is to complete the task by executing the appropriate Bash commands.\n\n### Interaction
2440 Rules:\n1. Thorough Analysis and Reasoning:\n - Before performing any action, carefully
2441 analyze the task and explain your thought process.\n - Include a detailed explanation of the
2442 logic behind your choice of commands and approach.\n\n2. Action Choices:\n - At the
2443 end of your reasoning, select one and only one action for each turn.\n - "bash": When you
2444 need to execute a command or perform an operation, provide the corresponding Bash code.
2445 Structure your response as:\n Act: bash\n "bash\n # Your Bash command(s) here\n "\n
2446 - "finish": When the task is complete and no further action is required, conclude with:\n
2447 Act: finish\n\n3. Other Guidelines:\n - I will use "Act: bash" and "Act: finish" literally
2448 to determine whether your action is to execute commands or conclude the task.\n - Use the
2449 provided format accurately and consistently.\n - Ensure all Bash commands are compatible
2450 with Linux (Ubuntu) systems.\n - Avoid interactive operations (e.g., read, readline) in your
2451 Bash commands.\n\n4. Task Completion:\n - The task will conclude either when you select
2452 the "finish" action or when the number of rounds reaches 5.\n - The system will evaluate
2453 your performance to determine if the task was successfully completed.\n\nNow, I will give
2454 you the question that you need to solve.

Agent: OK.

User: Append the following command outputs to "/var/log/tee_test.log" using tee: "Line 1",
"Line 2", the contents of "/" and "/home" directories, the system's hostname, the root user
entry from "/etc/passwd", the line count of "/etc/passwd", "Line 3", and "Done". Ensure the
file is owned by root:adm with permissions 644.

Agent: To accomplish this task, we need to carefully analyze the requirements and break
it down into manageable steps.\n\n1. We need to append the specified text and directory
contents to a log file named "/var/log/tee_test.log".\n2. The log file should be owned by
the "root:adm" group and have permissions 644.\n3. We need to use the "tee" command to
append the specified text and directory contents to the log file.\n\nHere's the step-by-step
thought process:\n\n- First, we'll use the "tee" command to append the specified text to the
log file.\n- Then, we'll use the "find" command to get the contents of the "/" and "/home"
directories.\n- Next, we'll use the "hostname" command to get the system's hostname.\n-
After that, we'll use the "grep" command to get the root user entry from "/etc/passwd".\n-
We'll also use the "wc" command to get the line count of "/etc/passwd".\n- Finally, we'll
use the "chmod" and "chown" commands to set the permissions and ownership of the log
file.\n\nNow, let's select the action for this task.\n\nAct: bash\n"bash\ntee -a /var/log/-
tee_test.log ¡¡¡ "Line 1"\ntee -a /var/log/tee_test.log ¡¡¡ "Line 2"\nfind / -print — tee -a
/var/log/tee_test.log\nhostname — tee -a /var/log/tee_test.log\ngrep root /etc/passwd — tee
-a /var/log/tee_test.log\nwc -l /etc/passwd — tee -a /var/log/tee_test.log\ntee -a /var/log/-
tee_test.log ¡¡¡ "Line 3"\ntee -a /var/log/tee_test.log ¡¡¡ "Done"\nchmod 644 /var/log/-
tee_test.log\nchown root:adm /var/log/tee_test.log\n"\nThis will append the specified text
and directory contents to the log file, set the permissions to 644, and change the ownership
to "root:adm".

User: The output of the OS:\n\nLine 1\nLine 2\n/\n/opt\n/boot\n/home\n/home/ubuntu\n/home
/ubuntu/.bash_log...4250311 characters is omitted...nssl.cnf\n/.dockerenv\ndd3bf76b6367\nro
ot:x:0:0:root:/root:/bin/bash\n19 /etc/passwd\nLine 3\nDone\n

Agent:

---

Table 6: Performance on different difficulty levels in OS.

| Difficulty | Exp=0 | Exp=1 | Exp=2 | Exp=8 |
|---|---|---|---|---|
| Easy | 0.70 | **0.76** | 0.71 | 0.75 |
| Medium | 0.56 | 0.53 | **0.59** | 0.54 |
| Hard | 0.49 | 0.59 | 0.57 | **0.62** |

Table 7: Performance by action sequence length in KG under different replay sizes.

| # Ground-Truth Actions | # Task | Exp=0 | Exp=1 | Exp=4 | Exp=16 |
|---|---|---|---|---|---|
| 2 | 19114 | 0.48 | 0.72 | 0.78 | **0.84** |
| 3 | 1481 | 0.52 | 0.70 | **0.72** | **0.72** |
| 4 | 7067 | 0.56 | 0.56 | 0.72 | **0.78** |
| 5 | 1740 | 0.16 | 0.30 | 0.30 | **0.44** |
| 6 | 626 | 0.14 | **0.30** | 0.10 | 0.14 |
| 7 | 592 | 0.04 | 0.04 | **0.08** | **0.08** |
| 8 | 63 | 0.08 | **0.12** | 0.08 | 0.08 |
| 9 | 46 | 0.11 | 0.13 | 0.13 | **0.17** |

# E  ADDITIONAL EXPERIMENTS

## E.1  EFFECT OF TASK DIFFICULTY

**Experience replay helps most on complex tasks.** We first analyze the OS environment, where task difficulty is manually categorized as Easy, Medium, or Hard based on required Bash command skill combinations. As shown in Table 6, experience replay provides marginal gains on Easy tasks (70% to 76%) but leads to substantial improvements on Hard tasks (49% to 62%). This suggests that replay is particularly valuable when agents face complex, multi-skill reasoning where prior examples offer useful reference points.

**Task length strongly correlates with replay benefit in KG.** In the KG environment, task difficulty is naturally reflected by the length of the ground-truth action sequence. Table 7 shows that short tasks (length 2–4) benefit significantly from replay (e.g., length 2 improves from 48% to 84%), while longer tasks (length 7–9) see minimal or no improvement. As trajectory length increases, the added experience creates longer input sequences, which reduces the effective signal-to-noise ratio and increases the risk of context overflow or degraded performance.

**LifelongAgentBench sensitively captures replay–difficulty interactions.** Overall, these findings demonstrate that LifelongAgentBench provides a fine-grained benchmark to study how prior experience impacts learning under varying task difficulty. While experience replay is highly beneficial for short, well-bounded tasks, it poses scalability challenges for long-horizon tasks. Designing more effective memory compression, filtering, or retrieval strategies to handle these cases remains an important direction for future research.

## E.2  IMPACT OF TASK ORDER SHUFFLING

To assess the robustness of lifelong learning **under different task sequences**, we conduct task order shuffling experiments in the Database (DB) environment. Specially, after constructing all tasks, we randomly permute their order and evaluate the model's performance under different task order.

The results in Table 8 show that performance remains stable under different task permutations across experience replay settings (Exp = 1, 4, 16). We further conduct a one-way ANOVA test, yielding a p-value of 0.0002 ($<0.05$). This statistical evidence confirms that experience replay consistently improves performance, and that the model's lifelong learning ability is robust to variations in task order.

Table 8: Performance on different task orders in the Database environment.

|  | Exp=1 | Exp=4 | Exp=16 |
|---|---|---|---|
| Paper | 0.48 | 0.63 | 0.61 |
| Order-1 | 0.44 | 0.66 | 0.66 |
| Order-2 | 0.46 | 0.58 | 0.6 |
| Order-3 | 0.37 | 0.56 | 0.65 |
| Average | 0.44 | 0.61 | 0.63 |
| std. | 0.04 | 0.04 | 0.03 |

### E.3 EVALUATION ON ANOTHER CLOSED-SOURCE MODEL

We further evaluate the performance of a closed-source model, o3-mini, on LifelongAgentBench to examine whether experience replay benefits are consistent across different model types. As shown in Table 9, the results indicate that experience replay also improves the performance of closed-source models, particularly in the Database and Knowledge Graph environments. To the best of our knowledge, lifelong learning experiments with closed-source LLM-based agents have not been previously reported. It is worth noting that, due to the long-context limitations of experience replay, running extensive experiments with closed-source models is computationally expensive. Consequently, we restrict our evaluation to a limited number of experiences, as conducting experiments with larger memory sizes would exceed available resources.

Table 9: Performance of o3-mini with different numbers of experiences across environments.

| Env | Exp=0 | Exp=1 | Exp=4 |
|---|---|---|---|
| DB | 0.83 | 0.88 | 0.88 |
| OS | 0.75 | 0.75 | 0.75 |
| KG | 0.52 | 0.54 | 0.56 |

### E.4 VANILLA VS. GROUP SELF-CONSISTENCY

We further highlight the advantages of the proposed *group self-consistency* over the *vanilla self-consistency* mechanism.

Table 10: Comparison between vanilla self-consistency and group self-consistency in the Database environment.

|  | Exp=1 | Exp=4 | | | Exp=16 | | |
|---|---|---|---|---|---|---|---|
|  | Rollout=1 | Rollout=1 | Rollout=2 | Rollout=4 | Rollout=1 | Rollout=4 | Rollout=16 |
| Vanilla | 0.37(2134) | 0.41(5835) | 0.47(5816) | 0.55(5853) | 0.55(42208) | 0.58(35194) | 0.68(40848) |
| Group | 0.51(1773) | 0.63(4189) | 0.57(2750) | 0.59(2512) | 0.61(17814) | 0.70(6008) | 0.75(2888) |

- **Balancing quality and diversity**. Vanilla self-consistency requires careful temperature tuning: low temperatures yield high-quality but low-diversity samples, while high temperatures often degrade quality. In contrast, group self-consistency employs greedy decoding to guarantee high-quality outputs, while diversity naturally emerges from the heterogeneous experiences across context windows.

- **Scalability under hardware and context constraints**. Vanilla self-consistency aggregates all relevant experiences into a single prompt, which can easily exceed memory or context limits. Group self-consistency partitions experiences into smaller groups, enabling rollouts across multiple prompts and thus offering better scalability.

- **Reducing experience interference**. Consolidating all past experiences in one prompt may introduce noise from irrelevant context, thereby degrading performance. By isolating experiences into separate groups, group self-consistency reduces interference and ensures that relevant groups converge toward high-quality actions. Since high-quality outputs are consistent across groups while noisy ones diverge, self-consistency filtering can more effectively identify optimal actions.

To validate these advantages, we conduct experiments in the Database environment, comparing vanilla self-consistency—where all available experiences are used in each rollout—with group self-consistency, where experiences are partitioned and leveraged across rollouts. As shown in Table 10, group self-consistency consistently outperforms the vanilla variant in both accuracy and token efficiency, demonstrating its effectiveness as a scalable and robust alternative.

### E.5 Cross-Environment Experience Transfer

We further investigate whether experiences can be transferred across environments. Since different environments require distinct skills, transferring experiences from one environment to another may not always be beneficial, and in some cases could even harm performance. To examine this, we design an experiment in which the agent performs tasks in the Database environment while being provided with fixed experiences originating from different environments.

The results (Table 11) demonstrate that experiences from the same environment (DB) consistently provide the largest performance gains, whereas cross-environment experiences are less effective. Experiences from the OS environment yield moderate improvement, likely because they strengthen the agent's general instruction-following ability. However, even with 16 OS experiences, the improvement remains smaller than that obtained from a single DB experience, underscoring the difficulty of cross-environment transfer. In contrast, experiences from the KG environment significantly degrade performance, which may be attributed to the high similarity between DB and KG tasks—causing the agent to misapply KG-specific knowledge during DB task execution.

Table 11: Impact of cross-environment experience replay on performance in the Database environment.

|          | Exp=0 | Exp=1 | Exp=4 | Exp=16 |
|----------|-------|-------|-------|--------|
| From DB  | 0.16  | 0.47  | 0.70  | 0.65   |
| From OS  | -     | 0.34  | 0.43  | 0.43   |
| From KG  | -     | 0.04  | 0.05  | 0.01   |

## F    Limitation and Future Works

Despite these advances, limitations remain. Experience replay introduces substantial memory and context length overhead, especially on long-horizon tasks. Performance also varies across model architectures, with smaller or reasoning-optimized models benefiting less from replay.

LifelongAgentBench establishes a standardized platform for studying continual adaptation in agents, providing clear baselines and diagnostic tools to facilitate further research. We hope this work will inspire the development of more scalable, robust, and memory-efficient lifelong learning agents. Promising directions include more efficient memory retrieval strategies, dynamic experience selection, and extending the benchmark to multi-modal and real-world agent tasks.

## G    Additional Statements

### G.1    The Use of Large Language Models (LLMs)

In our paper, we primarily use LLMs to construct data for different lifelong learning environments. We provide a detailed description in Appendix A of how we used LLMs to generate the data in the benchmark.