# OpenReview forum: "LifelongAgentBench: Evaluating LLM Agents as Lifelong Learners"
_ICLR.cc/2026/Conference — Submitted to ICLR 2026_

### Official Review · Reviewer_mS98 · 2025-10-24

**Soundness:** 2
**Presentation:** 2
**Contribution:** 2
**Rating:** 2
**Confidence:** 3

**Summary:**

This paper proposes LifelongAgentBench, the first unified benchmark framework specifically designed to evaluate the capabilities of LLM-based agents in lifelong learning scenarios. Unlike previous evaluation methods that treat agents as static systems, this framework emphasizes assessing an agent’s ability to accumulate, retain, and transfer knowledge within continuous, interdependent task sequences.

Main contributions include:
- Systematic analysis of the effects of experience replay: Identifies limitations of traditional approaches in LLM-based agents, such as interference from irrelevant information and context length constraints.
- Proposing the group self-consistency mechanism: Improves decision quality by grouping historical experiences and using voting, significantly alleviating memory and reasoning overhead issues.

**Strengths:**

- This work is the first to propose a benchmark specifically targeting the lifelong learning capability of LLM-based agents, with a novel problem definition that fills a gap in existing evaluation frameworks.
- The proposed grouped self-consistency mechanism represents an improvement over traditional experience replay methods, demonstrating methodological innovation.
- The work offers an extensive suite of well-defined and verifiable agent tasks, enabling performance evaluation and experiments.

**Weaknesses:**

There are flaws in the experimental aspect:
1. On line 054, table 1 only includes a few agent-related benchmarks for comparison. Examples like osworld and browsecomp were not taken into consideration.
2. On line 328, table 2 intends to express the effectiveness of replay, but it only uses one model.
3. Line 435, Table 3 only measured DB and KG. Additionally, the number of models used for DB and KG was different. If a model fails in KG, then DB should not be included either, as it has no significance.
4. On line 270, fig 3 is the only experiment that used a closed-source model. Why wasn't it presented in a table?

Overall, as a benchmark, it fails to provide sufficient evaluation results using both open-source and closed-source models. The types and quantities of models used in each experiment are very arbitrary. There was no appropriate ablation study for the proposed replay and vote methods.

**Questions:**

See the "Weaknesses" section.

---

### Official Review · Reviewer_PjtQ · 2025-10-28

**Soundness:** 2
**Presentation:** 1
**Contribution:** 1
**Rating:** 2
**Confidence:** 3

**Summary:**

This paper introduces LifelongAgentBench, a benchmark intended to evaluate the lifelong learning capabilities of  LLM agents. The authors posit that existing benchmarks fail to assess knowledge accumulation over time. The benchmark provides a sequence of skill-grounded tasks in three environments (Database, OS, Knowledge Graph). The paper's primary evaluation focuses on in-context experience replay (ER), finding that replaying relevant experiences is superior to replaying recent ones. It also proposes "group self-consistency", a voting method, to manage the context-length limitations of this replay strategy.

**Strengths:**

Important Problem: The paper's core motivation is strong. Evaluating the ability of agents to learn continuously is a critical, timely, and under-studied problem in the field of LLM-based agents.

Benchmark Artifact: The creation of a dedicated, open-source benchmark with containerized environments and automatic verification is a non-trivial engineering effort. This infrastructure could, in principle, be a useful tool for the community.

**Weaknesses:**

Unclear Definition of "Lifelong Learning": The paper fails to provide a precise and operational definition of lifelong learning. In Section 3, the problem formulation is presented as a generic sequential POMDP, which does not capture any distinctive characteristics of lifelong tasks. No explicit statement is given to clarify what “lifelong” means in this context, and how it affects the benchmark design.

Overstated Novelty and Weak Analysis: The claimed methodological contribution appears minor and is overstated. The concept of group self-consistency (Section 6.5) seems to be a straightforward rebranding of standard self-consistency methods (e.g., Wang et al., 2023), without a clear theoretical or methodological differentiation. The “systematic analysis” is limited, focusing only on the comparison between experience replay and group self-consistency, which does not provide a comprehensive understanding of the proposed approach in relation to the broader body of methods.

Poor Clarity and Presentation: The paper is difficult to follow due to unclear exposition of core concepts. Terminology such as "skill concurrency", "skill-grounded", "label verification" (is this equivalent to "label validation"?), "parallel execution", etc., is introduced without proper definitions or contextual examples. Figures and tables suffer from poor readability (extremely small font sizes; Figure 1 is overly cluttered). The paper does not convincingly justify what makes its task dependencies uniquely "lifelong", as similar setups could be replicated using existing benchmarks with replay-based agents. Table 1 lists differences, but it is unclear why these differences provide specific advantages under a lifelong learning scenario. Further explanation is necessary. Also, Table 1 is inconsistent with the “four key innovations” described later in the text, very confusing. Several citations are incorrect (e.g., VisualWebArena, AgentBench, wrong authors, wrong links), are they AI generated?

**Questions:**

I highly doubt that some major parts are written by llms without careful checking. I strongly recommend that the authors carefully review these paragraphs and thoroughly refine the wording to improve clarity and precision. Due to the poor readability of many parts, I may have overlooked some of the paper’s potential contributions. A significant improvement in writing quality would positively influence my evaluation and may result in a higher score.

---

### Official Review · Reviewer_eERx · 2025-10-29

**Soundness:** 2
**Presentation:** 1
**Contribution:** 1
**Rating:** 2
**Confidence:** 5

**Summary:**

This work introduces LifelongAgentBench, the first unified benchmark for evaluating LLM agents’ lifelong learning across databases, operating systems, and knowledge graphs. It features task dependency, verifiable labels, reproducibility, and modularity. Experiments show traditional experience replay is limited, while a grouped self-consistency mechanism boosts performance. Experience quality matters more than quantity, and model architecture and task difficulty strongly affect replay effectiveness.

**Strengths:**

1. The benchmark is highly reliable and flexible, easy to use, and readily extensible.
2. The grouped self-consistency mechanism effectively mitigates memory and inference overhead in large-scale experience replay.

**Weaknesses:**

1. The paper shows limited novelty. Among the four claimed innovations, Task Dependency is a common method for constructing tasks and does not clearly differ from prior work. Label Verifiability and Reproducibility are basic requirements for a benchmark, while Modularity relates to usability. Only Task Dependency contains some technical content, and the others cannot be considered true innovations.
2. The evaluation of lifelong learning is incomplete because it only considers rapid adaptation to new tasks and does not assess whether new experiences cause forgetting on previously learned tasks. LifelongAgentBench does not measure the impact of new experiences on old tasks.
3. The benchmark includes too few agent environments, which limits the generality of the conclusions.
4. Many tasks in the database and operating system environments are generated by DeepSeek-R1, making them synthetic and potentially misaligned with real-world human task distributions.
5. The paper defines lifelong learning narrowly, essentially by adding past experiences to the context, which resembles few-shot learning. Observed results, such as small gains for strong base models or performance improvement with more experiences, are common across tasks and not specific to agent settings.
6. The writing and focus of the paper are problematic because it emphasizes lifelong learning while devoting most of the content to engineering details rather than conceptual or methodological contributions.

**Questions:**

The conclusion mentions that adding experience to a high-performing base model yields little improvement, and can even be detrimental. What, then, are the challenges faced by such strong base models in lifelong learning, and how can they be addressed?

---

### Comment · Area_Chair_QiGV · 2025-11-28
**please respond to reviewer concerns promptly**

Dear Authors,

A quick reminder that the ICLR 2026 rebuttal window will close in less than a week. To ensure a fair and thorough evaluation, we encourage you to address the reviewers’ concerns as soon as possible.

Thank you for your prompt attention and for helping us keep the process efficient.

AC

---

### Meta-Review · Area_Chair_UV8K · 2026-01-07

**Summary:**

This paper presents LifelongAgentBench, a benchmark for testing whether LLM-based agents can improve over time across three environments (Database, Operating System, Knowledge Graph), with containerized setup and automatic checking. The benchmark infrastructure is useful, but the paper’s novelty feels overstated: items like verifiable labels, reproducibility, and modular design are expected for modern benchmarks. The paper also does not clearly define what “lifelong learning” means here, and most experiments reduce it to adding past experiences into the prompt, without measuring forgetting on earlier tasks. The experimental coverage across models/environments is insufficient, and the writing and citations need significant cleanup.

**Reviewer Concerns:**

no rebuttal

**Reviewer Scores:**

no rebuttal

---

### Decision · Program_Chairs · 2026-01-26

Reject